# Insights from a methylome-wide association study of antidepressant exposure

E. Davyson [1,2], X. Shen [1], F. Huider [3,4,5], M. J. Adams [1], K. Borges[1], D. L. McCartney [2], L. F. Barker [6], J. van Dongen [3,4,5,7], D. I. Boomsma [3,4,7], A. Weihs [8,9], H. J. Grabe [8,9], L. Kühn [8], A. Teumer [8,10], H. Völzke[10,11], T. Zhu[12,13], J. Kaprio [12], M. Ollikainen [12,13], F. S. David [14,15], S. Meinert[16,17], F. Stein[15,18], A. J. Forstner[14,19,20], U. Dannlowski[16], T. Kircher [15,18], A. Tapuc [21,22], D. Czamara [22], E. B. Binder [22], T. Brückl[22], A. S. F. Kwong[1,23], P. Yousefi [23,24,25], C. C. Y. Wong[26], L. Arseneault [26], H. L. Fisher [26,27], J. Mill [28], S. R. Cox [29], P. Redmond[29], T. C. Russ [1,30,31], E. J. C. G. van den Oord [32], K. A. Aberg [32], B. W. J. H. Penninx [33], R. E. Marioni [2], N. R. Wray [6,34] & A. M. McIntosh [1] ✉

This study tests the association of whole-blood DNA methylation and anti-depressant exposure in 16,531 individuals from Generation Scotland (GS), using self-report and prescription-derived measures. We identify 8 associations and a high concordance of results between self-report and prescription-derived measures. Sex-stratified analyses observe nominally significant increased effect estimates in females for four CpGs. There is observed enrichment for genes expressed in the Amygdala and annotated to synaptic vesicle membrane ontology. Two CpGs (cg15071067; *DGUOK-AS1* and cg26277237; *KANK1)* show correlation between DNA methylation with the time in treatment. There is a significant overlap in the top 1% of CpGs with another independent methylome-wide association study of antidepressant exposure. Finally, a methylation profile score trained on this sample shows a significant association with antidepressant exposure in a meta-analysis of eight independent external datasets. In this large investigation of antidepressant exposure and DNA methylation, we demonstrate robust associations which warrant further investigation to inform on the design of more effective and tolerated treatments for depression.

Major Depressive Disorder (MDD) is predicted to become the leading cause of disability worldwide by 2030[1], partly due to the limitations of current treatments[2]. Although antidepressants are commonly pre-scribed effective treatments[3], they prove to be ineffective in a high proportion of cases, with an estimated 40% of those presenting with MDD developing treatment-resistant depression[4,5]. Furthermore, many treatments are commonly accompanied by side effects, including weight changes, dizziness, fatigue and sexual dysfunction[2]. There is

a need for more effective and better-tolerated antidepressant treat-ments and to target existing treatments to those most likely to respond. Advances are hampered by poor mechanistic understanding of both MDD itself[6,7] and how currently prescribed antidepressants lead to therapeutic effects[8].

The mechanism of currently prescribed antidepressants is incom-pletely understood. Initial theories surmised that their therapeutic effects were primarily due to the inhibition of monoamine reuptake in

the synapse, leading to an increase in monoamine concentrations in the brain[9]. However, antidepressant treatment has a delayed onset for symptomatic improvement, which does not reflect the immediate effect on monoamine levels[10]. This casts doubt on the simple role of monoamines as a causal factor in MDD[7,8,11], although other experimental paradigms continue to suggest their importance[12]. Another prominent theory suggests that antidepressants exert their therapeutic effects by increasing brain-derived neurotrophic factor (BDNF), leading to synaptic remodelling[13] and enhanced neuronal plasticity[8,14]. A recent study found evidence of antidepressants binding directly the BDNF receptor (neurotrophic tyrosine kinase receptor 2;TRKB), proposing this as a potential mechanism of action independent of changes in monoamine concentrations[15]. Additionally, sex differences in MDD risk[16,17], antidepressant efficacy and side-effects are well-documented[18], which may reflect sex-specific differences in neuronal circuitry[19]. However, precise mechanisms of these differences are unclear.

DNAm, the addition of a methyl group at a cytosine-phosphate-guanine (CpG) site, regulates gene expression and impacts cellular function[20,21]. The evidence for the effect of antidepressants on DNA methylation (DNAm) is growing[22,23]. In vitro studies found that the antidepressant paroxetine interacted with DNA methyltransferase (DNMT), a key enzyme involved in DNAm[24]. Furthermore, studies of chronically stressed rodent models have found that stress-induced DNAm and behavioural changes are reversed through both anti-depressant treatment[25] and DNMT inhibitors[26]. In 2022, Barbu et al.[27] performed a methylome-wide association study (MWAS) of self-reported antidepressant exposure in a subset of participants in Generation Scotland (GS, $N = 6428$) and the Netherlands Twin Register (NTR, $N = 2449$), and identified altered DNAm near to genes involved in the innate immune response in those exposed to antidepressants[27]. As self-report measures may be unreliable due to volunteer recall bias, a poor understanding of the medication nosology, and non-disclosure[28–30], Barbu et al.[27] also performed an MWAS of anti-depressant exposure based on recorded antidepressant prescriptions in the last 12 months. However, this assumes continuous treatment, potentially overestimating exposure due to general low adherence to antidepressant medication[31]. The calculation of active treatment periods from consecutive prescribing events could provide a potentially more reliable identification of antidepressant exposure[32]. Here, adherence to the medication is assumed given regular prescription dispensations at an expected frequency (given their amount and dosage), rather than a singular prescribing event.

In our study, we build upon previous analyses by Barbu et al.[27] by analysing a larger sample of GS ($N = 16,531$), and by estimating active treatment periods from prescribing records to identify those exposed to antidepressants at DNAm measurement. First, an MWAS was performed on both the self-report and prescription-derived measures of antidepressant exposure. Second, the MWAS analyses were restricted to MDD cases only to assess potential confounding by MDD, and sex-stratified analyses were conducted to investigate any sex-specific effects. Third, functional follow-up analysis of differentially methylated CpG sites was performed. Fourth, we investigated the enrichment of top CpGs in GS and an independent MWAS conducted in the Netherlands Study of Depression and Anxiety (NESDA). Fifth, the relationship between time in treatment and DNAm at significant CpG sites was investigated. Finally, a methylation profile score (MPS) for self-report antidepressant exposure was trained in GS and tested for an association with antidepressant exposure in eight independent external datasets: Finn Twin Cohort (FTC), Study of Health in Pomerania (SHIP-Trend), Lothian Birth Cohort 1936 (LBC1936), FOR2107, NTR, Avon Longitudinal Study of Parents and Children (ALSPAC), Munich Antidepressant Response Study-Unipolar Depression Study (MARS-UniDep) and the Environmental Risk (E-Risk) Longitudinal Twin Study, alongside a prospective sample of GS: Stratifying Depression and Resilience Longitudinally (STRADL) (Fig. 1).

## Results

The demographics of both antidepressant exposure phenotypes in the GS are shown in Table 1. To assess demographic differences between the exposed and unexposed group, we used Welch's independent samples t-tests for continuous variables (age, BMI) and chi-squared tests for categorical variables (sex, smoking status and lifetime MDD status).

In both self-report and prescription-derived measures, the anti-depressant exposed group were significantly older ($t_{self-report}(1970) = 10.6$, $p_{self-report} = 1.01 \times 10^{-25}$, $t_{prescription-derived}(1180) = 7.35$, $p_{prescription-derived} = 3.63 \times 10^{-13}$) and had significantly higher BMI measurements ($t_{self-report}(1700) = 11.4$, $p_{self-report} = 5.45 \times 10^{-29}$, $t_{prescription-derived}(980) = 9.82$, $p_{prescription-derived} = 8.69 \times 10^{-22}$). Additionally, both phenotypes showed that the antidepressant-exposed group had significantly different proportions of current, former and never smokers ($\lambda^2_{self-report}(2) = 160$, $p_{self-report} = 1.69 \times 10^{-35}$, $\lambda^2_{prescription-derived}(2) = 114$, $p_{prescription-derived} = 1.49 \times 10^{-25}$), a significantly higher proportion of females ($\lambda^2_{self-report}(1) = 214$, $p_{self-report} = 1.53 \times 10^{-48}$, $\lambda^2_{prescription-derived}(1) = 129$, $p_{prescription-derived} = 7.62 \times 10^{-30}$) and significantly higher proportion of those with lifetime-MDD ($\lambda^2_{self-report}(1) = 2170$, $p_{self-report} < 1 \times 10^{-320}$, $\lambda^2_{prescription-derived}(1) = 1450$, $p_{prescription-derived} = 2.72 \times 10^{-318}$).

### Methylome-wide association studies

The self-report MWAS (Fig. 2A, Table 2) and prescription-derived MWAS (Fig. 2B, Supplementary Table 1) identified seven and four hypermethylated CpGs respectively, in those exposed to anti-depressants (Supplementary Figs. 1, 2). The effect estimates from all CpGs in both analyses were significantly correlated ($R = 0.54$, $p < 2.2 \times 10^{-16}$) (Supplementary Fig. 3).

In the MDD-subgroup self-report MWAS, only cg08527546 exhibited a significant association with antidepressant exposure ($\beta = 0.050$, $p = 3.57 \times 10^{-8}$); no CpGs were significantly associated in the MDD-subgroup prescription-derived MWAS (Supplementary Table 2, Supplementary Fig. 4). For both phenotypes, there was a significant correlation between CpG effect estimates in the full and MDD-subgroup analyses ($R_{self-report} = 0.57$, $p_{self-report} < 2.2 \times 10^{-16}$, $R_{prescription} = 0.43$, $p_{prescription} < 2.2 \times 10^{-16}$) (Supplementary Fig. 5). Notably, restricting the analyses to MDD cases resulted in an average 2.5-fold and 2.1-fold increase in the self-report and prescription-derived effect sizes of the significant CpGs respectively (Fig. 2C).

In the sex-stratified analyses, two CpGs were significantly associated with self-report antidepressant exposure in females, cg26277237 ($\beta = 0.030$, $p = 1.77 \times 10^{-10}$) and cg02183564 ($\beta = 0.023$, $p = 4.29 \times 10^{-9}$) (Supplementary Table 3) and there were no significant associations with antidepressant exposure in males (Supplementary Fig. 6). The effect sizes of CpGs in the male-only and female-only analyses were significantly correlated ($R = 0.039$, $p < 2.2 \times 10^{-16}$) (Supplementary Fig. 7). Of the eight CpG sites significantly associated with antidepressant exposure in the overall analysis, all demonstrated a larger effect size in females (Supplementary Figs. 7-8). Four of these sites showed nominally significant sex differences (cg26277237; $p = 0.0081$, cg02183564; $p = 0.0132$, cg08907118; $p = 0.0335$, cg03222540; $p = 0.0305$) (Supplementary Data 1).

### Differentially methylated regions

There were 719,506 candidate regions considered in the analysis. The self-report MWAS had one significant DMR ($\beta = 0.096$, $p_{adj} = 4.98 \times 10^{-3}$) (Supplementary Data 2), consisting of two CpGs (cg01964004 and cg15071067), which maps to deoxyguanosine kinase antisense RNA1 (DGUOK-AS1) (Supplementary Fig. 9). The prescription-derived MWAS identified no significant DMRs (Supplementary Data 3).

### Functional Annotation

The most significant 100 CpGs from the self-report and prescription-derived MWAS were annotated to 77 and 83 genes respectively

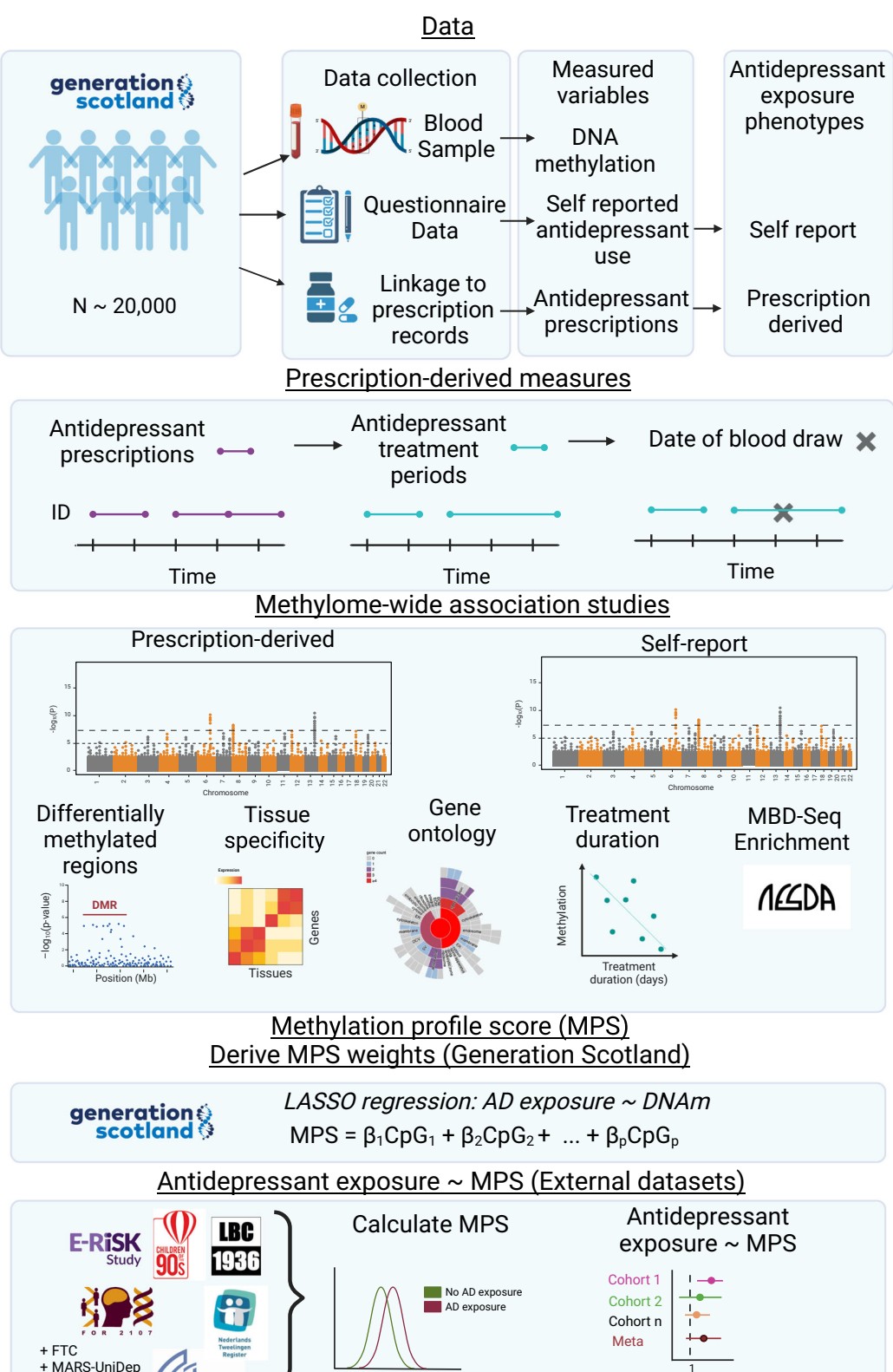

(Supplementary Data 4-7). There was a significant overlap between both the CpG-sets ($N_{overlap} = 17$, $p = 1.95 \times 10^{-48}$, Supplementary Fig. 10) and the gene-sets ($N_{overlap} = 16$, $p = 1.3 \times 10^{-25}$, Supplementary Fig. 11). The self-report gene-set was significantly enriched for the genes expressed in the amygdala ($p_{FDR} = 0.043$,

Supplementary Data 8, Supplementary Fig. 12), whilst the prescription-derived gene-set had no significant enrichment (Supplementary Data 9, Supplementary Fig. 13). There was significant enrichment of the self-report gene-set in GO:0008021; "synaptic vesicle membrane" ($p_{FDR} = 0.030$, Supplementary Figs. 14-16, Supplementary

**Fig. 1 | Study design of investigating antidepressant exposure and DNA methylation.** Data: Participants in Generation Scotland (GS) provided blood samples from which DNA methylation was measured. Their antidepressant exposure status was measured using both self-report questionnaires and prescription-derived measures. Prescription-derived measures: Repeated regular prescriptions over time (X axis) for antidepressants (purple bars) are merged to form active antidepressant treatment periods (blue bars). Individuals in an active treatment period at the time of blood sample (black cross) are classed as antidepressant exposed. Methylome-wide association studies: A methylome-wide association study (MWAS), subsequent regional analysis and functional annotation was performed for both measures of antidepressant exposure. Additionally, an enrichment analysis was done using MBD-Sequencing data in the Netherlands Depression and Anxiety (NESDA) cohort for the self-report antidepressant exposure. Methylation Profile Score: Weights for a methylation profile score (MPS) of self-reported antidepressant exposure was calculated in GS using a least absolute shrinkage and selection operator (LASSO) model. Eight independent datasets then tested the association of this MPS with self-reported antidepressant exposure. Created in BioRender. Davyson, E. (2024) https://BioRender.com/s43g100.

Tables 4-5, Supplementary Data 10-11). There was no significant enrichment for any GO:Biological Processes pathways tested for either gene set (Supplementary Figs. 17-18, Supplementary Data 12-13).

### Enrichment analysis: Netherlands Study of Depression and Anxiety

We tested whether top findings from our self-reported MWAS on all participants were also more likely to be among the top findings in an independent MWAS of antidepressant exposure in the Netherlands Depression and Anxiety (NESDA) cohort[33]. Notably, all participants in this cohort had a recent (~6 months) MDD DSM-IV diagnosis, obtained using the Composite International Diagnostic Interview, including single-episode and recurrent MDD (Supplementary Methods). DNAm was assayed on whole-blood samples from 812 MDD patients, 398 of which reported using one or more antidepressants, using methyl-CG binding domain sequencing (MBD-Seq)[34–36]. Data quality control and the MWAS was performed using RaMWAS[37] (Supplementary Methods). Enrichment tests were performed to assess the top findings from GS and NESDA MWAS. The enrichment analyses were performed using the 'shiftR' R package[38] that performs circular permutations to account for having correlated methylation levels between CpG sites. For both MWAS', CpGs were filtered to those which showed a concordant direction of effect and three thresholds (0.1, 0.5 and 1%) were used to define top (by $p$-value) findings. We corrected for this "multiple testing" by using the same thresholds in the permutations and selecting the most significant result from each permutation to generate the "empirical" null distribution. Results suggested a small (OR: 1.39) but significant ($P < 0.042$) enrichment between the top 1% of results from both MWAS' (Supplementary Data 14).

### Methylation differences by time in treatment

To investigate the relationship of DNAm at significantly associated CpGs ($n = 8$) with the length of antidepressant exposure, a two-tailed Spearman correlation test was performed between the DNAm levels and time in current treatment for those within a treatment period ($N = 863$). Two probes, cg15071067 (*DGUOK-AS1*) and cg26277237 (*KANK1*), showed a significant correlation between methylation and time in treatment (cg15071067:$\rho$= 0.085, $p = 0.012$, cg26277237: $\rho$= 0.087, $p = 0.011$) (Supplementary Figs. 19-21, Supplementary Table 6), with the same direction of effect found in the MWAS.

### Methylation profile score

There were 212 CpGs identified by the LASSO regression model (Supplementary Figs. 22-24, Supplementary Data 15), used to calculate the MPS in external cohorts (Fig. 3A, Supplementary Figs. 25-34). All cohorts bar one (NTR) showed a positive relationship between antidepressant MPS and antidepressant exposure ($\beta_{FTC} = 0.156$, $\beta_{SHIP-Trend} = 0.134$, $\beta_{STRADL} = 0.149$, $\beta_{LBC1936} = 0.228$, $\beta_{FOR2107} = 0.349$, $\beta_{MARS} = 0.263$, $\beta_{ALSPAC} = 0.170$, $\beta_{ERISK} = 0.342$, $\beta_{NTR} = -0.031$) (Supplementary Fig. 35, Supplementary Table 7). Nagelkerke's pseudo $R^2$ estimates ranged from $1.11 \times 10^{-3}$ (NTR) to 0.03 (LBC1936) (Fig. 3B). The random-effects meta-analysis (Fig. 3C) found a significant association between antidepressant exposure and the MPS (pooled $\beta$ [95%CI]: 0.196 [0.105, 0.288],

**Table 1 | Demographics and structured clinical interview of the DSM (SCID) diagnoses of antidepressant exposed and unexposed individuals using the prescription-derived and self-reported antidepressant exposure phenotypes in Generation Scotland**

|  | Prescription-derived phenotype | | Self-report phenotype | |
|---|---|---|---|---|
|  | Unexposed | Exposed | Unexposed | Exposed |
| N | 7090 | 861 | 15028 | 1508 |
| Age: M (SD) | 47.6 (15.2) | 51.1 (12.7) | 46.6 (14.9) | 50.2 (12.4) |
| BMI: M (SD) | 26.4 (4.78) | 28.5 (6.11) | 26.4 (5.01) | 28.2 (6.05) |
| Sex |  |  |  |  |
| Female: N (%) | 3747 (53) | 631 (73) | 8556 (57) | 1154 (77) |
| Male: N (%) | 3343 (47) | 230 (27) | 6467(43) | 354 (23) |
| Smoking behaviours |  |  |  |  |
| Current: N (%) | 963 (14) | 214 (25) | 2448 (16) | 402 (27) |
| Former: N (%) | 1942 (27) | 273 (32) | 4183 (28) | 481 (32) |
| Never: N (%) | 3913 (55) | 328 (38) | 8114 (54) | 576 (38) |
| Pack years: M (SD) | 6.3 (13.1) | 12 (17.1) | 6.55 (13.3) | 11.6 (16.9) |
| Structured Clinical Interview for DSM IV (SCID) |  |  |  |  |
| No Major Disorder: N (%) | 6514 (92) | 373 (43) | 12704 (85) | 556 (37) |
| Bipolar Disorder | 9 (0.1) | 9 (1) | 29 (0.2) | 23 (1.5) |
| Single Episode MDD N (%) | 281 (4) | 137 (16) | 880 (6) | 260 (17) |
| Recurrent MDD: N (%) | 131 (2) | 243 (28) | 622 (4) | 506 (34) |
| MDD cases (Recurrent & Single Episode): N (%) | 412 (6) | 380 (44) | 1502 (10) | 766 (51) |

*M* Mean, *SD* Standard Deviation, *MDD* Major Depressive Disorder.

$p < 1 \times 10^{-4}$), with low heterogeneity between studies (I² [95%CI] = 0% [0, 64.8%]) (Supplementary Table 8).

## Discussion

This study presents a large investigation of antidepressant exposure and the methylome[39]. There was evidence of hypermethylation at eight individual CpGs and a singular region on Chromosome 2 (BP: 74196550-74196572) in those exposed to antidepressants using both self-report and prescription-derived measures[32]. Sex-stratified analyses indicated larger effect estimates in females compared to males. Functional annotation found that genes annotated to the top 100 differentially methylated CpGs in the self-report MWAS were significantly enriched in both a synaptic vesicle membrane gene-ontology and for genes expressed in the amygdala. Preliminary analysis indicated a significant correlation between the time in treatment and methylation at two probes (cg15071067; *DGUOK-AS1* and cg26277237; *KANK1*). Furthermore, there was significant enrichment in the top 1% of findings from an independent MWAS using MBD-Seq profiling. Finally, an MPS trained on GS data, demonstrated a robust association with antidepressant exposure in a

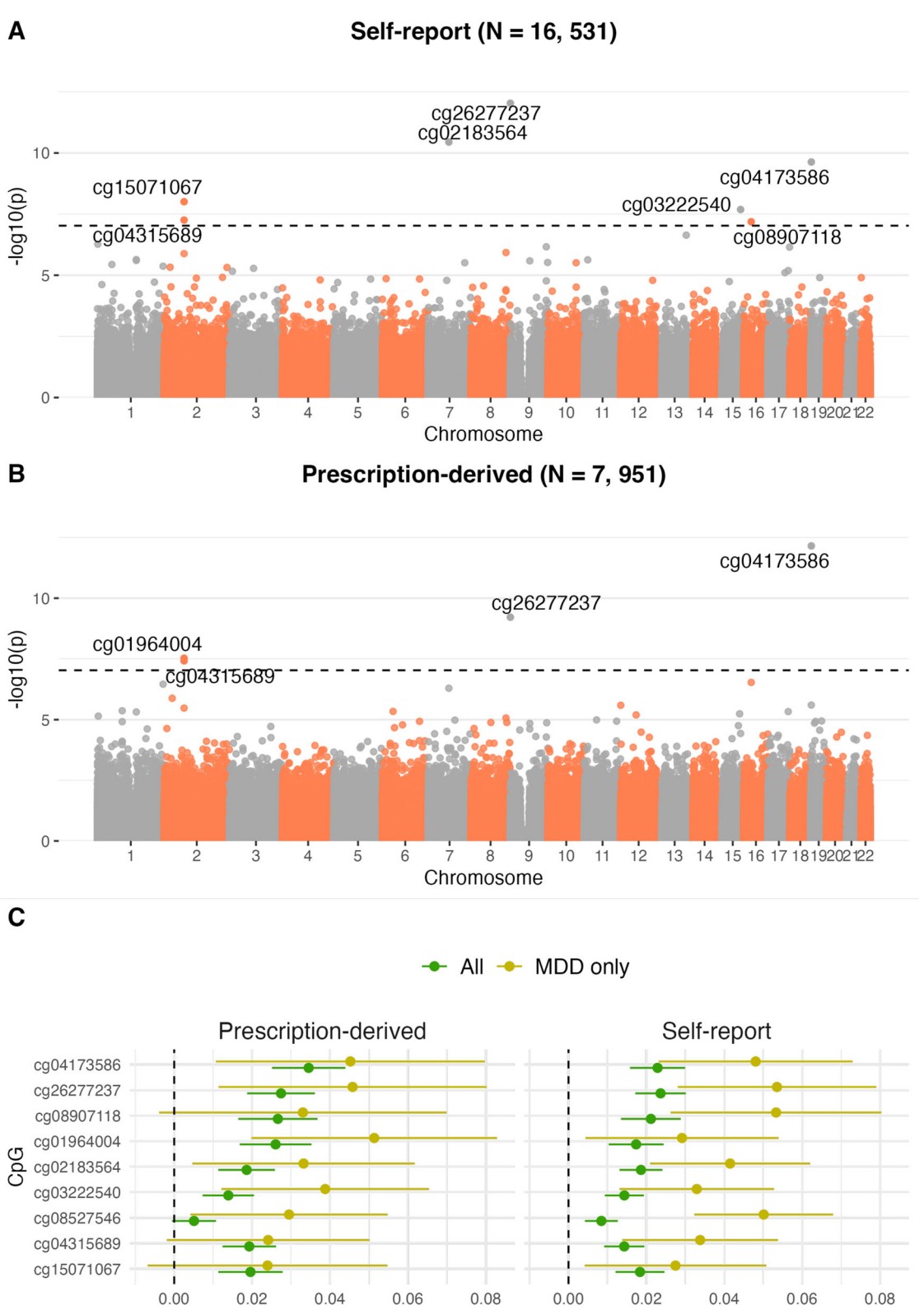

**Fig. 2 | Methylome-wide association studies (MWAS) of self-reported and prescription-derived antidepressant exposure.** Manhattan plots of the MWAS of self-report (**A**) and prescription-derived (**B**) antidepressant exposure using a Mixed-linear-model Omics-based Analysis (MOA) model. Significance was assessed using the *p*-value threshold $9.42 \times 10^{-8}$, as recommended for case-control MWAS analyses[67]. **C** The MOA MWAS standardised effect sizes and 95% confidence-intervals (effect estimate +/- 1.96*Standard Error) for associated CpGs ($p < 9.42 \times 10^{-8}$) for the full sample ($N_{\text{self-report}}$ = 16,531, $N_{\text{prescription-derived}}$ = 7951) and MDD-subgroup sample ($N_{\text{self-report}}$ = 2268, $N_{\text{prescription-derived}}$ = 792). Data are presented as the MWAS effect estimates +/- the 95% confidence intervals. Effect sizes represent a per-1 standard-deviation increase in CpG methylation M-values.

**Table 2 | Eight CpGs associated with self-reported (SR) and/or prescription-derived (PD) antidepressant use**

| Probe | Chr | BP | Gene | Significant in SR / PD MWAS | $\beta_{SR}$ | $SE_{SR}$ | $P_{SR}$ | $\beta_{PD}$ | $SE_{PD}$ | $P_{PD}$ | Other traits associated with the CpG (EWAS catalog)[79] |
|---|---|---|---|---|---|---|---|---|---|---|---|
| cg26277237 | 9 | 631910 | KANK1 | SR + PD | 0.024 | 0.0033 | $9.3 \times 10^{-13}$ | 0.027 | 0.0044 | $6.1 \times 10^{-10}$ | C-reactive protein levels[80] (basic model), Chronic kidney disease (basic model)[81], Type 2 diabetes (basic model)[81], Estimated glomerular filtration rate[82] |
| cg04173586 | 19 | 2167496 | DOT1L | SR + PD | 0.023 | 0.0036 | $2.35 \times 10^{-10}$ | 0.034 | 0.0048 | $7 \times 10^{-13}$ | Alcohol consumption[83], COVID-19 severity (basic model)[81], Sex[84], Age[85], Alzheimer's disease Braak stage[86], Estimated glomerular filtration rate[82], Schizophrenia[87] |
| cg08907118 | 16 | 27482516 | GTF3C1 | SR only | 0.021 | 0.0039 | $6.46 \times 10^{-8}$ | 0.027 | 0.0052 | $3.0 \times 10^{-7}$ | C-reactive protein levels[80], Chronic pain (basic model)[81], Type 2 diabetes (basic model)[81], Sex[84], Age[85] |
| cg02183564 | 7 | 76874892 | CCDC146 | SR only | 0.019 | 0.0028 | $3.59 \times 10^{-11}$ | 0.019 | 0.0037 | $5.17 \times 10^{-7}$ | Birthweight[85], C-reactive protein levels (basic model)[80], Gestational age[88], Chronic pain (basic model)[81], Type 2 diabetes (basic model)[81], Age[85] |
| cg15071067 | 2 | 74196550 | DGUOK-AS1 | SR-only | 0.018 | 0.0032 | $9.84 \times 10^{-9}$ | 0.020 | 0.0042 | $3.3 \times 10^{-6}$ | C-reactive protein levels (basic model)[80], Incident chronic pain (basic model)[81], Incident type 2 diabetes (basic & fully adjusted model)[81], Chronic pain (self-report)[81], Type 2 diabetes (basic model)[81] |
| cg01964004 | 2 | 74196572 | DGUOK-AS1 | PD only | 0.017 | 0.0036 | $1.3 \times 10^{-6}$ | 0.026 | 0.0047 | $3.0 \times 10^{-8}$ | C-reactive protein levels (basic model)[80], Incident type 2 diabetes (basic & fully adjusted model)[81], Chronic pain (basic model)[81], Ischemic heart disease (basic model)[81], Type 2 diabetes (basic model)[81] |
| cg03222540 | 15 | 90575080 | ZNF710 | SR only | 0.014 | 0.002 | $2.05 \times 10^{-8}$ | 0.014 | 0.0034 | $3.7 \times 10^{-7}$ | C-reactive protein levels (basic model)[80], Chronic kidney disease (basic & fully adjusted model)[81], Type 2 diabetes (basic model)[81] |
| cg04315689 | 2 | 74198896 | DGUOK-AS1 | SR + PD | 0.014 | 0.0026 | $5.56 \times 10^{-8}$ | 0.019 | 0.0035 | $3.85 \times 10^{-8}$ | C-reactive protein levels (basic model)[80], Incident type 2 diabetes (basic model)[81], Chronic kidney disease (basic & fully adjusted model)[81], Chronic pain (basic model)[81], Type 2 diabetes (basic model)[81] |

The effect estimates ($\beta$), standard-errors (SE) and P-values (P) for the self-report ($_{SR}$) and prescription-derived ($_{PD}$) MWAS, calculated using a Mixed-linear-model Omics-based Analysis (MOA) model. The EWAS catalog was searched using the 'ewascatalog' R package[79] for other studies (n > 1000) which report a significant CpG-trait association, accessed on 17/03/2024.

meta-analysis of eight external cohorts and a prospective wave of data from GS, indicating the generalisability of our findings.

The CpG with the highest significance and the largest effect size, cg26277237, mapped to KN motif and ankyrin repeat domains 1 (*KANK1*), and was previously reported by Barbu et al.[27] on a smaller sample of GS. *KANK1* facilitates the formation of the actin cytoskeleton and has an active role in neurite outgrowth and neurodevelopment[40]. A meta-analysis of copy-number variant association studies found a significant duplication in *KANK1* in those with five different neurodevelopmental disorders, including MDD[41]. The DMR analysis indicated antidepressant exposure is also significantly associated with hypermethylation near *DGUOK-AS1*, a long non-coding RNA (lncRNA). *DGUOK-AS1* has an inhibitory role on the expression of a nearby gene *DGUOK*[42], which encodes a mitochondrial enzyme involved in the production of mitochondrial DNA[42], and has previously been implicated as a risk gene in schizophrenia[43] and Alzheimer disease[44]. A recent review reported evidence that antidepressants do influence mitochondrial function, although the effects are heterogeneous between different types of antidepressants, independent of their current classification[45,46]. Functional validation of the associations of antidepressant exposure with DNAm at *KANK1* and *DGUOK-AS1* would strengthen our findings. The integration of additional multi-omic data alongside in-vitro experiments could further assess the impact of these associations on biologically relevant processes, such as neuroplasticity.

Our results show broadly consistent findings between self-report and prescription-derived measures. However, self-report measures showed stronger signal in downstream functional annotation analyses. Due to the strong correlation between effect estimates and several overlapping significant signals, this is likely due to the increased sample size and power in the self-report cohort (+8580 individuals). The top CpGs in the self-report MWAS were significantly enriched for genes expressed in the amygdala, an important component of emotional brain circuits, specifically in regulating fear and stress responses[47]. Genomic studies of MDD have shown enrichment in neural synaptic pathways[48] and brain regions in the meso-limbic system, including the prefrontal cortex, nucleus accumbens, hippocampus and the amygdala[49]. Several meta-analyses have found evidence of amygdala hyperreactivity in those with MDD[50–52], while other studies have demonstrated the amygdala response to stress can be dampened through neuroplastic processes following antidepressant treatment[53] or cognitive behavioural therapy[54]. Furthermore, a recent systematic review reported sex-specific differences in amygdala activity and grey matter volume in MDD[16]. Notably, our sex-stratified analyses found nominally significant sex differences in the DNAm-antidepressant exposure associations at four significant CpGs, with larger effects observed in females. Although there is no clear consensus regarding sex differences in antidepressant efficacy, reports have found that women generally respond better to selective serotonin reuptake inhibitors (SSRIs) than men[19]. The nominally significant sex-differences in DNAm associations with antidepressant exposure and the significant enrichment of genes in the amygdala observed in this study highlight the potential role of the amygdala in mediating sex-specific responses to antidepressants. Future studies could investigate sex-specific DNAm-profiles of antidepressant exposure and their functional impact on the amygdala using functional imaging data.

There are several strengths of this study. The comparison of self-report and prescription-derived measures is valuable to the research community. Self-reported measures are often cheaper and easier to obtain in large-scale cohort studies[31]. Equally, the methods used in this study to derive of medication exposure using prescription records could enable passive data collection, enabling more generalisable analyses on whole populations outside of biases which influence participation in biobank cohorts[55]. Furthermore, the MDD-subgroup analysis indicates that the hypermethylation associated with antidepressant use is not

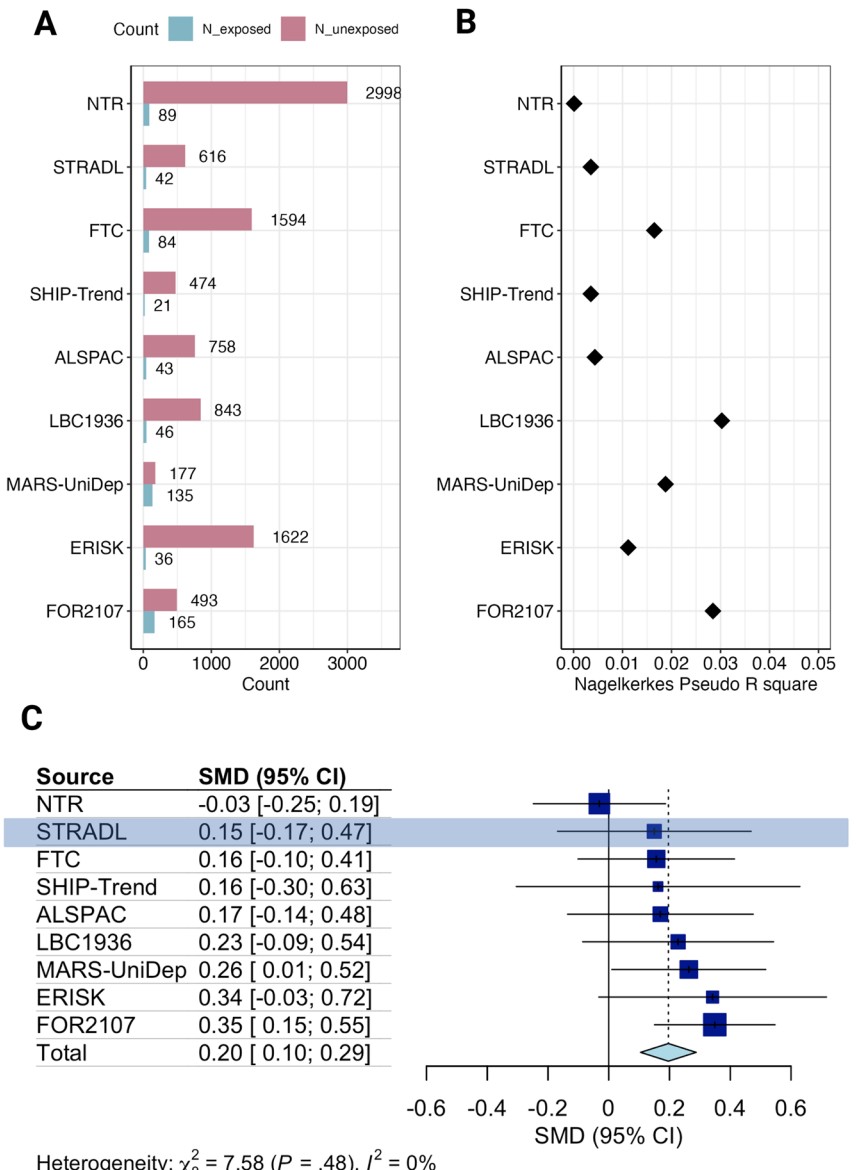

**Fig. 3 | An antidepressant exposure methylation profile score (MPS) and antidepressant exposure in external cohorts. A** The number of participants exposed and unexposed to antidepressants in each cohort ($N_{NTR} = 3087$, $N_{STRADL} = 658$, $N_{FTC} = 1678$, $N_{SHIP-TREND} = 495$, $N_{ALSPAC} = 801$, $N_{LBC1936} = 889$, $N_{MARS-UniDep} = 312$, $N_{ERISK} = 1658$, $N_{FOR2107} = 658$). **B** Nagelkerke's pseudo $R^2$, representing an estimate of how much variance in the antidepressant exposure outcome that is explained by the MPS in each cohort. **C** The effect estimates and 95% confidence intervals (effect estimate +/− 1.96*Standard Error) of MPS - antidepressant exposure association in each cohort, using either a generalised linear model (FOR2107 and ALSPAC), a generalised linear mixed model (SHIP-Trend, LBC1936, MARS-UniDep, STRADL and E-Risk) or a generalised estimation equation model (FTC and NTR). The square size = study weight in the random-effects meta-analysis. The pooled effect estimate interval, calculated from a random-effects meta-analysis, is represented by the blue diamond ($N_{pooled}=10,236$). Created in BioRender. Davyson, E. (2024) https://BioRender.com/q70l792.

primarily driven by MDD indication. Additionally, the performance of the GS-trained MPS in discriminating antidepressant exposure across eight external datasets, alongside the significant enrichment of top findings with an independent MWAS, demonstrates that this may be a generalisable biomarker indicative of antidepressant exposure.

This study has various limitations. Firstly, both measures of antidepressant exposure do not discriminate between antidepressant drugs, classes, or dosages. However, we anticipate the opportunity to investigate more medication-specific effects on the methylome using prescription-linkage data as biobanks increase in size. Secondly, all the cohorts used primarily consist of European ancestry. It is paramount that this analysis is conducted in non-European ancestral groups to further verify our findings and disentangle any ancestry-specific effects[56–58]. Thirdly, our epidemiological analyses do not adjust for various other potential confounders, such as comorbidities and concomitant treatments. However, adjusting for all possible confounds such as these may bias findings due to collinearity and collider bias. Fourthly, by design, this epidemiological study cannot directly address causality between antidepressant exposure and DNAm. The integration of DNAm analysis into randomised controlled trials of antidepressants is important to establish the exact nature of the association and to inform potential new targets for antidepressant therapy. Finally, this study does not examine recently approved rapid-acting antidepressants, such as ketamine. Future research could build on this by exploring the associations between DNAm and rapid-acting antidepressants, while comparing these effects with those of slower-acting antidepressants discussed in this study.

This study indicates that antidepressant exposure is associated with hypermethylation at *DGUOK-AS1* and *KANK1*, which have roles in mitochondrial metabolism and neurite outgrowth respectively. Future research should include more cohorts of non-European ancestry, alongside the incorporation of DNAm in randomised trials of antidepressants to establish causality. If replicated, targeting of these genes could inform the design of more effective and better tolerated treatments for depression.

## Methods

### Ethics declaration

All components of Generation Scotland: The Scottish Family Health Study (GS) received ethical approval from the NHS Tayside Committee on Medical Research Ethics (REC Reference Number: 05/S1401/89). Generation Scotland has also been granted Research Tissue Bank status by the East of Scotland Research Ethics Service (REC Reference Number: 20-ES-0021), providing generic ethical approval for a wide range of uses within medical research. All participants included in the current study provided informed consent for the use of their data for biomedical research. The research presented in this study complies with these ethical regulations. The study protocol for the Netherlands Study of Anxiety and Depression (NESDA) was approved centrally by the Ethical Review Board of the VU University Medical Centre and subsequently by local review boards of each participating centre (METC number 2003-183). After full verbal and written information about the study, written informed consent was obtained from all participants at the start of baseline assessment. Ethics approvals have been granted for multiple studies concerning the Finnish Twin Cohort (FTC) twins by the ethics committees of Helsinki University Central Hospital (113/E3/2001, 249/E5/2001, 346/E0/05, 270/13/03/01/2008, and 154/13/03/00/2011) with the last one on the transfer of biological samples to the THL Biobank in 2018 (HU51179912017). Participants in the FTC were given information on the study procedures and of freedom to participate or to decline at any point in both oral and written form. Informed consent was obtained upon the contact with the study subjects before new questionnaire information was collected, and when clinical investigations were undertaken with sampling of biological material. The Study of Health in Pomerania (SHIP) received ethical approval from the ethics Committee of the University Medicine Greifswald, Germany (BB 39/08). All participants provided written informed consent before any assessment and/or sampling took place. FOR2107 received ethical approval from the ethics committees of the Medical Faculties, University of Marburg (AZ: 07/14) and University of Münster (2014-422-b-S). Participants provided written consent before any assessment and/or sampling took place. The Netherlands Twin Register was approved by the Central Ethics Committee on Research Involving Human Subjects of the VU university Medical Centre, Amsterdam, an Institutional Review Board certified by the U.S Office of Human Research Protections (IRB number IRB00002991 under Federal-wide Assurance-FWA00017598; IRB/institute codes, NTR 03-180). Informed consent was obtained from all participants. The Munich Antidepressant Response (MARS) study was approved by the local Ethics Committee of the Ludwig Maximilians University, Munich, Germany (318/00; 244/01) and carried out in accordance with the latest version of the Declaration of Helsinki. All participants provided written consent after the study protocol and potential risks were explained. The UniPolar Depression (UniDep) study was approved by the Bavarian State Medical Association (BLAEK) (01217) and written informed consent was obtained from all subjects. The Lothian Birth Cohort 1936 (LBC1936) received ethical approval from the Multicentre Research Ethics Committee for Scotland (baseline, MREC/01/0/56), the Lothian Research Ethics Committee (age 70, LREC/2003/2/29), and the Scotland A Research Ethics Committee (ages 73, 76, 79, 07/MRE00/58). All participants provided written

informed consent. Ethical approval for the Avon Longitudinal Study of Parents and Children (ALSPAC) was obtained from the ALSPAC Ethics and Law Committee and the Local Research Ethics Committees under proposal B3818. Consent for biological samples has been collected in accordance with the Human Tissue Act (2004). Each study phase of the Environmental risk (E-risk) Longitudinal Twin Study received ethical approval from the Joint South London and Maudsley and Institute of Psychiatry Research Ethics Committee (NRES 1997/122). Parents gave informed consent, and twins gave assent between 5 and 12 years and then informed consent at age 18. All components of the Stratifying Resilience and Depression Longitudinally (STRADL) received formal national ethical approval from the NHS Tayside committee on research ethics (reference 14/SS/0039).

### Generation Scotland

Generation Scotland: The Scottish Family Health Study (GS) is a family-based cohort study (N ~ 24,000), investigating the genetic and environmental factors influencing health within Scotland (Supplementary Methods)[59,60]. Data collection including biological sampling occurred between February 2006 and March 2011.

### Methylation data

DNAm was profiled from baseline blood samples using the Illumina MethylationEPIC array for 19,390 individuals. Methylation typing was performed in 4 distinct sets between 2017 and 2021. Sets 1, 2 and 4 included related individuals within and between each set while all individuals in set 3 were unrelated to each other and to individuals in set 1 (genetic relationship matrix (GRM) threshold <0.05). Following quality control (QC) (Supplementary Methods), the sample sizes were Set 1: 5087, Set 2: 459, Set 3: 4450 and Set 4: 8873 individuals. The sets were combined and dasen normalisation was performed across all individuals[61]. In total, 752,741 probes and 18,869 individuals passed QC. Beta-values were transformed to M-values using the 'beta2M()' function in the 'lumi' R package[62], and standardised using the 'std-probe' flag in the OSCA software[63].

### Antidepressant exposure phenotypes

Prescription-derived antidepressant exposure was measured using antidepressant prescriptions from Public Health Scotland (Supplementary Methods), according to the British National Formulary (BNF) paragraph code '040303', which largely corresponds to Anatomical Therapeutic Chemical (ATC) subclass 'N06A' (Supplementary Table 9) ($N_{prescriptions}$ = 174,454, $N_{people}$ = 7544) (Supplementary Table 10, Supplementary Figs. 36-43). After removing ambiguous prescriptions ($N_{prescriptions}$ = 5484, $N_{people}$ = 171) (Supplementary Methods, Supplementary Table 11), active treatment periods were defined by consecutive dispensing of antidepressant medications (Supplementary Methods, Supplementary Fig. 44-46). Those who had their blood-draw appointment ≥7 days after treatment initiation or <7 days after the end of a treatment period were defined as exposed ($N_{exposed}$ = 861) (Supplementary Fig. 47). Those with no antidepressant prescriptions on record were defined as unexposed ($N_{unexposed}$ = 7090).

Self-reported antidepressant exposure was derived from questionnaires sent 1–2 weeks before venepuncture (Supplementary Methods, Supplementary Table 12, Supplementary Fig. 48). Those who did/did not self-report antidepressant use were defined as exposed and unexposed respectively ($N_{exposed}$ = 1508, $N_{unexposed}$ = 15,023). Out of 6473 individuals with both self-report and prescription-derived phenotypes, 6355 exhibited concordant classification of antidepressant exposure (Supplementary Fig. 49). The MDD-only phenotypes were derived by stratifying the samples to those with a lifetime MDD diagnosis, ascertained by the Structured Clinical Interview of the Diagnostic and Statistical Manual, version IV (SCID)[64] (prescription-derived: $N_{exposed}$ = 380, $N_{unexposed}$ = 412, self-report: $N_{exposed}$ = 766, $N_{unexposed}$ = 1502) (Supplementary Methods, Supplementary Fig. 50).

## Methylome-wide association study

The MWAS were performed using a Mixed-linear-model Omics-based Analysis (MOA) in the OSCA software[63]. To account for relatedness within GS, each phenotype was regressed on a GRM[65] using the Best Linear Unbiased Prediction (BLUP) tool in GCTA software[66] ('reml-pred-rand' flag). The residuals were entered into a MOA model, which included a methylation omics-relatedness matrix as a random effect and age, sex, AHRR probe (cg05575921) M-values to proxy for smoking status, and predicted monocyte and lymphocyte cell proportions as fixed effects. Statistical significance was assessed using the $p$-value threshold $9.42 \times 10^{-8}$, as recommended for case-control MWAS analyses[67]. Effect sizes represent a per-1 standard-deviation increase in CpG methylation M-values. For any significant CpGs, we searched the EWAS catalog to assess their associations with other traits in the literature. For sex-stratified analyses, we divided our participants by self-reported sex (male/female) and used the same MOA model without sex as a covariate.

Differentially methylated regions (DMRs) were identified using the 'dmrff' R package[68], which performs an inverse-variance-weighted meta-analysis of MWAS beta and standard-error estimates per region, adjusting for estimate uncertainty and the correlational structure between probes. Candidate DMRs are identified as sets ( >2) of CpGs <= 500 bp apart with nominal significance ($P < 0.05$) and consistent effect direction. DMRs achieving Bonferroni-corrected $p$-value < 0.05 were considered statistically significant.

## Functional annotation

Gene-sets for both MWAS were collated by annotating the top 100 CpGs (by $p$-value) by the Infinium MethylationEPIC BeadChip database[69]. Hypergeometric tests, using '*phyper()*', were used to assess the overlap of CpGs and gene-sets from both analyses. The background set consisted of CpGs and genes on or annotated to the EPIC array (Supplementary Data 16). The 'GENE2FUNC' analysis in functional mapping and annotation web-tool (FUMA) was used to assess enrichment of both gene-sets across 54 specific tissues in the Genotype-Tissue Expression (GTEx)[70,71] Project. Both gene-sets were tested for enrichment in synapse-related GO terms using the SynGo web tool[72] and for enrichment in GO: Biological Processes gene-sets (20 <ngenes <500) in the msigdbr database[73], using the 'gsameth()' function from 'missMethyl' R package[74] (Supplementary Data 17).

## Methylation profile score

A least absolute shrinkage and selection operator (LASSO) penalised regression model was applied using 'cv.biglasso()' from the 'glmnet' R package[75] on the GS sample to generate weights for a MPS of antidepressant exposure. First, the self-report phenotype was regressed against the GRM (using BLUP) to account for relatedness, and then on age, sex, AHRR probe M-values as a proxy for smoking status, batch, and white blood cell (monocyte and lymphocyte) proportions. The extracted residuals were used as the dependent variable and the standardised CpG sites on both EPIC and 450 K Illumina arrays ($N = 365,912$) were included as independent variables. Ten-fold cross-validation was applied, and the mixing parameter was set to 1. The non-zero weighted CpGs identified in the LASSO model were used to calculate a weighted-sum MPS in external datasets (FTC, SHIP-Trend, FOR2107, NTR, LBC1936, ALSPAC, MARS-UniDep and E-Risk, $N_{total} = 9578$, $N_{exposed} = 619$) and a prospective sample of GS (STRADL, $N_{total} = 658$, $N_{exposed} = 42$) (Supplementary Methods). The association between antidepressant exposure and the MPS was assessed using; generalised linear mixed models, generalised linear models and generalised estimation equations, depending on the cohort's population structure (i.e., twin studies vs unrelated participants) and DNAm pre-processing (Supplementary Methods). In-depth information about each cohort's sample, DNAm pre-processing and associational model can be found in the supplementary information (Supplementary Methods, Supplementary Data 18-20). All models have antidepressant exposure as the outcome and include age at blood sampling, sex, batch (where applicable), white blood cell proportions/counts and M-values at the AHRR probe as covariates. A random-effects meta-analysis using a DerSimonian-Laird estimator was performed to assess the overall association between the MPS and antidepressant exposure, using the 'meta' R package[76].

## Statistics and reproducibility

No statistical method was used to predetermine the sample size in either the self-report or prescription-derived MWAS analysis. Sample size was determined by the availability of eligible individuals (with DNA methylation and phenotypic information available) in each cohort. For all the MWAS', samples that failed quality control checks (Supplementary Methods) were excluded from the analysis. For the MDD-subgroup analyses, individuals who fulfilled criteria for bipolar disorder in the SCID assessment ($N_{prescription-derived} = 18$, $N_{self-report} = 52$) were discarded. For external cohorts which utilised inpatient and outpatient samples (FOR2107, MARS and UniDep) there were further exclusionary criteria. In FOR2107, individuals with a bipolar, schizophrenia or schizoaffective disorder diagnosis were excluded. In MARS, individuals were excluded if they displayed hypomanic symptoms, a diagnosis of alcohol dependence, a history of illicit drug use or having depressive symptoms secondary to another medical or neurological condition. In UniDep, individuals were excluded if they had manic or hypomanic episodes, mood incongruent psychotic symptoms, a lifetime diagnosis of intravenous drug abuse and depressive symptoms which are secondary to a substance abuse disorder or due to medical illness or medication. The study design did not involve randomization as this is an observational study utilizing data from population-based cohorts. The investigators were not blinded to the exposure or outcome variables as this was an observational study with no experimental allocation of treatments.

## Reporting summary

Further information on research design is available in the Nature Portfolio Reporting Summary linked to this article.

## Data availability

The data generated in this study (MWAS summary statistics, downstream functional results and weights in the MPS) have been deposited in a Zenodo repository (14203229 [https://doi.org/10.5281/zenodo.14203229])[77]. Additionally, the source data for main figures and all Supplementary Figs. which are not sharing individual-level data are provided in this repository. The raw data from all cohorts (Generation Scotland, Stratifying Anxiety and Depression Longitudinally, Netherlands Study of Anxiety and Depression, Finnish Twin Cohort, Study of Health in Pomerania, FOR2107, Netherlands Twin Register, Munich Antidepressant Response Study, UniPolar Depression Study, Avon Longitudinal Study of Parents and Children, E-risk Longitudinal Twin Study and Lothian Birth Cohorts) are not publicly available due to them containing information that could compromise participant consent and confidentiality. Access information for each cohort can be found below. Generation Scotland is run as a resource for the research community. Requests to use the Resource are made from: Academic collaborators: employees who are party to the Generation Scotland Collaboration Agreement, or researchers or employees of an academic institution or the NHS. Commercial organisations: specific arrangements have been defined to allow commercial organisations to access Generation Scotland resources. Data can be obtained from the data owners. Instructions for accessing Generation Scotland data can be found here: https://genscot.ed.ac.uk/for-researchers/access; the GS Access Request Form can be downloaded from this site. Completed request forms must be sent to access@generationscotland.org to be approved by the Generation Scotland Access Committee. Upon

submission, applications are reviewed within 6-8 weeks. For any further correspondence and material requests please contact genscot@ed.ac.uk. The Netherlands Study of Anxiety and Depression (NESDA) is run as a resource for the research community and is open to data-use requests from bona fide international researchers. Instructions for gaining access to NESDA can be found here: https://www.nesda.nl/nesda/wp-content/uploads/2024/09/NESDA_policy_data_access.pdf. Applications involve submitting a research proposal, including specific research questions, methodology and proposed statistical analysis, to the NESDA management committee (nesda@amsterdamumc.nl). Data access forms can be downloaded here: www.nesda.nl. The review process by the management committee for data access may take up to 6 weeks. E-Risk is run as a resource for the research community and is free to access by researchers from all over the world who are based at universities or research institutions. E-Risk operates a managed access process to protect the privacy of the participants. Instructions for accessing The Environmental-Risk Longitudinal Twin Study (E-Risk) data can be found here: https://eriskstudy.com/data-access. In brief following the reading of a data-sharing protocol, a concept paper will need to be submitted outlining the proposed analyses and required variables via this link: https://redcap.link/ERiskConceptPaperForm for consideration by the E-Risk Steering Committee. A decision will usually be made within 1 month. Once approved, the concept paper will be made public and the data sent securely to those requiring access within 1-2 months after a formal data use agreement has been signed. For any questions regarding the data available and the access process, please email eriskstudy@kcl.ac.uk. The data of the Study of Health in Pomerania study cannot be made publicly available due to the informed consent of the study participants, but it can be accessed through a data application form available at https://transfer.ship-med.uni-greifswald.de/ for researchers who meet the criteria for access to confidential data. In detail, access to the data requires a proposal submitted to the University Medicine Greifswald represented by the steering committee of the Research Network Community Medicine (FVCM), which meets once a month. A successful data application provides data usage permission for three years at maximum and needs to be extended afterwards if needed. In accordance with the Finnish Biobank Act, the data used in the analysis from the Finnish Twin Cohort (FTC) is deposited in the Biobank of the Finnish Institute for Health and Welfare (https://thl.fi/en/research-and-development/thl-biobank/for-researchers). It is available to researchers from academia and companies after written application and following the relevant Finnish legislation. To ensure the protection of privacy and compliance with national data protection legislation, a data use/transfer agreement is needed, the content and specific clauses of which will depend on the nature of the requested data. Allow 2-4 weeks for initial response from the Biobank. Avon Longitudinal Study of Parents and Children (ALSPAC) is run as a resource for the research community. Instructions for accessing ALSPAC data can be found here: https://www.bristol.ac.uk/alspac/researchers/access/. A research proposal must be submitted via the research proposal system for consideration by the ALSPAC Executive Committee. For any questions regarding accessing data or samples please email alspac-data@bristol.ac.uk (data) or bbl-info@bristol.ac.uk (samples). Approval may take up to two weeks. Lothian Birth Cohort 1936 (LBC1936) is run as a resource for the research community which actively collaborates with research experts in the UK and internationally. Instructions for how to access LBC1936 data can be found here: https://lothian-birth-cohorts.ed.ac.uk/data-access-collaboration. In brief, identify the variables you will require for the analysis from the LBC1936 data dictionary (which can be downloaded from the link). Following this, prepare a data request form including the provisional title of the study, the principle researcher and institution and the brief rationale for the study, research method and main variables. Email the LBC request form to Professor Simon Cox (simon.cox@ed.ac.uk), the director of Lothian Birth Cohorts for approval from the study investigator team. Approval may take up to 4 weeks. Data will be shared on the basis of a Data or Material Transfer Agreement between provider and recipient institutions. The Netherlands Twin Register (NTR) is run as a resource for the research community and is open to data-use requests from bona fide international researchers. Information on accessing NTR data can be found here: https://ntr-data-request.psy.vu.nl/. To submit a data sharing request, complete a data sharing request form (https://ntr-data-request.psy.vu.nl/DSR-forms.html) and send it to NTR Data management team (ntr.datamanagement.fgb@vu.nl), which will check the request for feasibility and completeness and pass on the request to the data access committee (DAC) for approval. The review process by the DAC may take up to 4 weeks. The raw data collected in the FOR2107 study is not openly accessible to protect participant consent and confidentiality. Nevertheless, the FOR2107 study serves as a valuable resource for the global research community. It is accessible, in principle, to all scientific researchers affiliated with non-commercial research organizations worldwide. Researchers seeking access to the study's data must submit a formal research proposal. This proposal should outline the specific research questions, methodology, and planned statistical analyses. Applications are reviewed by the principal investigators of FOR2107, Professors Tilo Kircher (tilo.kircher@staff.uni-marburg.de) and Udo Dannlowski (udo.dannlowski@uni-muenster.de), within 6–8 weeks of submission. The raw data collected in the Munich Antidepressant Response Study-Unipolar Depression Study (MARS-UniDep) is not publicly available but can be shared with researchers upon request. Access to the data can be requested in a data transfer agreement, please contact Darina Czamara (darina@psych.mpg.de). Only research questions related to psychiatric disorders can be addressed directly. The expected time frame for response to access requests is 2 weeks. Data access will be granted until the end of the requested project, that is upon publication of the related manuscript.

## Code availability

All code and analysis scripts are available on GitHub (https://github.com/Elladavyson/Antidepressant_MWAS) and as a linked Zenodo object (14185886 [https://doi.org/10.5281/zenodo.14185886])[78]. Furthermore, a capsule of the code processing the MWAS summary statistics, downstream analyses and the meta-analysis results is provided at Code Ocean (https://doi.org/10.24433/CO.6842127.v2).

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

## Acknowledgements

This work has made use of the resources provided by the Edinburgh Compute and Data Facility (ECDF, 2023) (http://www.ecdf.ed.ac.uk/). E.D. was supported by the United Kingdom Research and Innovation (grant EP/S02431X/1), UKRI Centre for Doctoral Training in Biomedical AI at the University of Edinburgh, School of Informatics. For the purpose of open access, the author has applied a creative commons attribution (CC BY) licence to any author accepted manuscript version arising. A.M.M. is supported by Wellcome (226770/Z/22/Z, 220857/Z/20/Z), UK Research and Innovation awards (Grant No. MR/W014386/1, MR/Z000548/1, MR/Z503563/1) and by European Union Horizon 2020 funding (Grant No. 847776). X.S. is supported by Wellcome (220857/Z/20/Z and 226770/Z/22/Z) and a UK Research and Innovation award (Grant no. MR/Z50354X/1). M.J.A. is supported by Wellcome (220857/Z/20/Z). F.H. is supported by the Netherlands Organization for Scientific Research (NWO 480-15-001/674) and Biobanking and Biomolecular Resources Research Infrastructure (BBMRI-NL: 184.021.007; 184.033.111) J.V.D. is supported by the Biobanking and Biomolecular Resources Research Infrastructure

(BBMRI-NL: 184.021.007). D.I.B. is supported by the Royal Dutch Academy for Arts and Science (KNAW) Academy Professor Award (PAH/6635); PI the Netherlands Organization for Scientific Research (NWO 480-15-001/674) and CoPI: Biobanking and Biomolecular Resources Research Infrastructure (BBMRI-NL: 184.021.007; 184.033.111). J.K. is supported by the Academy of Finland (grants 100499, 205585, 118555, 141054, 264146, 308248, 265240, 263278) and the Sigrid Juselius Foundation (grant #1057). M.O. is supported by the Academy of Finland (grants 328685, 307339, 297908 and 251316) and the Sigrid Juselius Foundation (grant #1105). A.S.F.K. is supported by the Wellcome Trust Early Career Award (227063/Z/23/Z). P.Y. is supported by National Institute for Health and Care Research (NIHR) Bristol Biomedical Research Centre and the Medical Research Council Integrative Epidemiology Unit at the University of Bristol (MC_UU_00032/4, MC_UU_00032/6). The views expressed are those of the authors and not necessarily those of the NIHR or the Department of Health and Social Care. C.C.Y.W. is supported by the National Institute for Health and Care Research (NIHR) Maudsley Biomedical Research Centre at South London and Maudsley NHS Foundation Trust and King's College London [NIHR203318]. H.L.F. is supported by the Economic and Social Research Council (ESRC) Centre for Society and Mental Health at King's College London [ES/S012567/1]. The views expressed are those of the authors and not necessarily those of the NIHR, the Department of Health and Social Care, the ESRC, or King's College London. J.M. has received support from the National Institute for Health and Care Research (NIHR) Exeter Biomedical Research Centre (BRC, NIHR203320). The views expressed are those of the author(s) and not necessarily those of the NIHR or the Department of Health and Social Care. S.R.C. is supported by the Biotechnology and Biological Sciences Research Council and the Economic and Social Research Council [BB/W008793/1], Age UK [Disconnected Mind project], the Medical Research Council [MR/R024065/1], the Milton Damerel Trust [Lothian Birth Cohorts], Sir Henry Dale Fellowship jointly funded by the Wellcome Trust and the Royal Society (221890/Z/20/Z). T.C.R. is supported by the Biotechnology and Biological Sciences Research Council and the Economic and Social Research Council [BB/W008793/1] and Age UK [Disconnected Mind project]. E.J.C.G.v.d.O and K.A.A. are supported by the National Institute of Mental Health (NIMH: R01MH124981). N.R.W. is supported by the Australian National Health and Medical Research Council (NHMRC: 1173790, 1113400). Generation Scotland (GS) & Stratifying Resilience and Depression Longitudinally (STRADL): We thank the participants in Generation Scotland for making this research possible. We also acknowledge the team at Generation Scotland for collecting and preparing the data for analyses. This work was supported by 3 Wellcome Trust grants (220857/Z/20/Z, 226770/Z/22/Z and 104036/Z/14/Z [AMM]). Funding from the Biotechnology and Biological Sciences Research Council and Medical Research Council (Grant Nos. MR/X003434/1 and MR/W014386/1) and the European Union (Grant agreement 847776) is also gratefully acknowledged. The Psychiatric Genomics Consortium (including AMM) has received major funding from the U.S. National Institute of Mental Health (Grant No. 5 U01MH109528-03). Netherlands Study of Anxiety and Depression (NESDA): The NESDA study is supported by the Geestkracht program of the Netherlands Organization for Health Research and Development (Zon-Mw, grant number 10–000-1002) and the participating institutions (VU University Medical Center, Leiden University Medical Center, University Medical Center Groningen. The current methylation project was supported by grant R01MH099110 from the National Institute of Mental Health. The sponsors had no role in the design and conduct of the study; collection, management, analysis, and interpretation of the data; preparation, review, or approval of the manuscript; or decision to submit the manuscript for publication. Finnish Twin Cohort (FTC): Phenotype and genotype data collection in FT12 and FT16 studies of the Finnish twin cohort has been supported by the Wellcome Trust Sanger Institute, the Broad Institute, ENGAGE – European Network for Genetic and Genomic Epidemiology, FP7-HEALTH-

F4-2007, grant agreement number 201413, National Institute of Alcohol Abuse and Alcoholism (grants AA-12502, AA-00145, and AA-09203 to R J Rose; AA15416 and K02AA018755 to D M Dick; R01AA015416 to J Salvatore) and the Academy of Finland (grants 100499, 205585, 118555, 141054, 264146, 308248, 265240, 263278 to J.K., 328685, 307339, 297908 and 251316 to M.O., and Centre of Excellence in Complex Disease Genetics grants 312073, 336823, and 352792 to J.K.), NIH/NHLBI (grant HL104125 to X Wang), and the Sigrid Juselius Foundation to M.O and J.K. Study of Health in Pomerania (SHIP-Trend): SHIP is part of the Community Medicine Research net of the University of Greifswald which is funded by the Federal Ministry of Education and Research (01ZZ9603, 01ZZ0103, and 01ZZ0403), the Ministry of Cultural Affairs and the Social Ministry of the Federal State of Mecklenburg-West Pomerania. DNA methylation data have been supported by the DZHK (grant 81×3400104). The University of Greifswald is a member of the Caché Campus program of the InterSystems GmbH. FOR2107: We are deeply indebted to all study participants and staff. A list of acknowledgments can be found here: www.for2107.de/acknowledgements. The German multicenter consortium "Neurobiology of Affective Disorders. A translational perspective on brain structure and function" is funded by the German Research Foundation (Research Unit FOR2107). Principal investigators are Tilo Kircher (speaker FOR2107, DFG grant numbers KI588/14-1, KI588/14-2, KI588/20-1, KI588/22-1, KI 588/15-1, KI 588/17-1), Udo Dannlowski (co-speaker FOR2107; DA1151/5-1, DA1151/5-2, DA1151/6-1), Axel Krug (KR3822/5-1, KR3822/7-2), Igor Nenadic (NE2254/1-2, NE2254/2-1, NE2254/3-1, NE2254/4-1), Carsten Konrad (KO4291/3-1), Marcella Rietschel (RI 908/11-1, RI 908/11-2), Markus Nöthen (NO 246/10-1, NO 246/10-2), Stephanie Witt (WI 3439/3-1, WI 3439/3-2). This work was funded in part by the consortium grant Trajectories of Affective Disorders from the German Research Foundation (DFG) SFB/TRR 393 (project grant no 521379614). Biosamples and corresponding data were sampled, processed and stored in the Marburg Biobank CBBMR. Netherlands Twin Register (NTR): We warmly thank all twin families of the Netherlands Twin Register who make this research possible. This work was supported by the Royal Dutch Academy for Arts and Science (KNAW) Academy Professor Award (PAH/6635) to D.I.B.; the Netherlands Organization for Scientific Research (NWO 480-15-001/674) and Biobanking and Biomolecular Resources Research Infrastructure (BBMRI-NL: 184.021.007; 184.033.111). Munich Antidepressant Response Study / UniPolar Depression Study (MARS-UniDep): We would like to thank all contributors to the research project including physicians, psychologists, study nurses, researchers and research assistants, and of course patients of the hospital of the Max Planck Institute of Psychiatry in Munich and psychiatric hospitals in Augsburg and Ingolstadt. The MARS cohort was sponsored by the Max Planck Society. The UniDep cohort was funded by the Bavarian Ministry of Commerce and by the Federal Ministry of Education and Research in the framework of the National Genome Research Network, Foerderkennzeichen 01GS0481 and the Bavarian Ministry of Commerce. DNA methylation analysis of a subset of both cohorts was financed by ERA-NET NEURON. Avon Longitudinal Study of Parents and Children (ALSPAC): We thank all the families who took part in this study, the midwives for their help in recruiting them, and the whole ALSPAC team, which includes interviewers, computer and laboratory technicians, clerical workers, research scientists, volunteers, managers, receptionists and nurses. The UK Medical Research Council and Wellcome (Grant Ref: 217065/Z/19/Z) and the University of Bristol provide core support for ALSPAC. A comprehensive list of grants funding is available on the ALSPAC website (http://www.bristol.ac.uk/alspac/external/documents/grant-acknowledgements.pdf). This publication is the work of the authors and E.D., P.Y. & A.S.F.K. will serve as guarantors for the ALSPAC contents of this paper. E-risk Longitudinal Twin Study (E-risk): We are grateful to the E-Risk study mothers and fathers, the twins, and the twins' teachers for their participation. Our thanks to Professors Terrie Moffitt and Avshalom Caspi, the founders of the E-Risk study, and to the E-Risk team for their dedication, hard work,

and insights. The E-Risk Study is funded by grants from the UK Medical Research Council [G1002190; MR/X010791/1]. Additional support was provided by the US National Institute of Child Health and Human Development [HD077482] and the Jacobs Foundation. Generation of DNA methylation data in E-Risk was supported by a Senior Investigator Award from the American Asthma Foundation (AAF) to J.M. High-performance computing was supported by an MRC Clinical Infrastructure award (MR/M008924/1) and we acknowledge the contribution and use of the CREATE high-performance computing cluster at King's College London (King's College London (2022). King's Computational Research, Engineering and Technology Environment (CREATE), https://doi.org/10.18742/rnvf-m076.) Lothian Birth Cohort 1936 (LBC1936): The LBC1936 is supported by the Biotechnology and Biological Sciences Research Council, and the Economic and Social Research Council [BB/W008793/1], Age UK (Disconnected Mind project), the Medical Research Council [G0701120, G1001245, MR/M013111/1, MR/R024065/1], the Milton Damerel Trust, and the University of Edinburgh. Methylation typing of was supported by Centre for Cognitive Ageing and Cognitive Epidemiology (Pilot Fund award), Age UK, The Wellcome Trust Institutional Strategic Support Fund, The University of Edinburgh, and The University of Queensland.

## Author contributions

E.D., A.M.M., R.E.M., and N.R.W. were responsible for the conception and the design of the study. E.D. conducted the primary data analyses,the MPS-meta-analysis and was primary author of the manuscript. X.S., M.J.A., D.L.M., and L.F.B., advised on methodology and data analysis. K.B. contributed to the derivation of prescription-derived antidepressant phenotypes. F.H., J.V.D., and D.B., ran the out-of-sample analysis in the Netherlands Twin Register. A.W., H.J.G., L.K., A.Teumer., and H.V., ran the out-of-sample analysis in Study of Health in Pomerania. T.Z., J.K., and M.O., ran the out-of-sample analysis in the Finnish Twin Cohort. F.S.D, S.M., F.S, A.J.F., U.D., and T.K., ran the out-of-sample analysis in FOR2107. A.Tapuc., D.C., E.B.B., and T.B ran the out-of-sample analysis in the Munich Antidepressant Response Study / UniPolar Depression Study. A.S.F.K., and P.Y., ran the out-of-sample analysis in the Avon Longitudinal Study of Parents and Children. C.C.Y.W., ran the out-of-sample analysis in the E-Risk Longitudinal Twin Study. L.A., H.L.F., and J.M., secured funding, oversaw data collection and the derivation of variables for the E-Risk Longitudinal Twin Study. S.R.C., P.R., and T.C.R., provided guidance and expertise on running the out-of-sample analysis in the Lothian Birth Cohorts. E.J.C.G.v.d.O, K.A.A., and B.W.J.H.P., ran the enrichment analysis in NESDA. A.M.M, R.E.M, N.R.W, X.S., and M.J.A. co-supervised the project. All authors read and approved the final manuscript.

## Competing interests

R.E.M. is an advisor to the Epigenetic Clock Development Foundation and Optima Partners. D.L.M. was employed by Optima Partners Ltd in a part-time capacity. H.J.G. has received travel grants and speaker honoraria from Fresenius Medical Care, Neuraxpharm, Servier, Indorsia and Janssen Cilag, as well as research funding from Fresenius Medical Care. H.J.G. had personal contracts approved by the university administration for speaker honoraria and one IIT with Fresenius Medical Care. T.K. received unrestricted educational grants from Servier, Janssen, Recordati, Aristo, Otsuka, neuraxpharm. B.W.J.H.P. has received research funding (not related to the current paper) from Boehringer Ingelheim and Jansen Research. All other authors report no biomedical financial interests or potential conflicts of interest.

## Additional information

[1]Division of Psychiatry, Centre for Clinical Brain Sciences, University of Edinburgh, Edinburgh, UK. [2]Institute of Genetics and Cancer, University of Edinburgh, Edinburgh, UK. [3]Complex Trait Genetics, Center of Neurogenomics and Cognitive Research, Vrije Universiteit Amsterdam, Amsterdam, The Netherlands. [4]Amsterdam Public Health Research Institute, Amsterdam, The Netherlands. [5]Department of Biological Psychiatry, Vrije Universiteit Amsterdam, Amsterdam, The Netherlands. [6]Institute for Molecular Bioscience, University of Queensland, Brisbane, QLD 4072, Australia. [7]Amsterdam Reproduction & Development, Research Institute, Amsterdam, The Netherlands. [8]Department of Psychiatry and Psychotherapy, University Medicine Greifswald, 17475 Greifswald, Germany. [9]German Center for Neurodegenerative Diseases (DZNE), Site Rostock/Greifswald, 17489 Greifswald, Germany. [10]German Centre for Cardiovascular Research (DZHK), Partner Site Greifswald, 17489 Greifswald, Germany. [11]Department SHIP/Clinical-Epidemiological Research, Institute for Community Medicine, University Medicine Greifswald, 17475 Greifswald, Germany. [12]Institute for Molecular Medicine Finland FIMM, HiLIFE, University of Helsinki, Helsinki, Finland. [13]Minerva Foundation Institute for Medical Research, Helsinki, Finland. [14]Institute of Human Genetics, University of Bonn, School of Medicine & University Hospital Bonn, Bonn, Germany. [15]Department of Psychiatry and Psychotherapy, University of Marburg, Marburg, Germany. [16]Institute for Translational Psychiatry, University of Münster, Münster, Germany. [17]Institute for Translational Neuroscience, University of Münster, Münster, Germany. [18]Center for Mind, Brain and Behavior, University of Marburg, Marburg, Germany. [19]Institute of Neuroscience and Medicine (INM-1), Research Center Jülich, Jülich, Germany. [20]Center for Human Genetics, University of Marburg, Marburg, Germany. [21]Max Planck School of Cognition, Leipzig, Germany. [22]Max-Planck-Institute of Psychiatry, Department Genes and Environment, Munich, Germany. [23]Medical Research Council Integrative Epidemiology Unit at the University of

Bristol, University of Bristol, Bristol, UK. [24]Population Health Science, Bristol Medical School, University of Bristol, Bristol, UK. [25]NIHR Bristol Biomedical Research Centre, University Hospitals Bristol and Weston NHS Foundation Trust and University of Bristol, Bristol, UK. [26]Social, Genetic & Developmental Psychiatry Centre, Institute of Psychiatry, Psychology & Neuroscience, King's College London, London, UK. [27]ESRC Centre for Society and Mental Health, King's College London, London, UK. [28]Department of Clinical & Biomedical Sciences, University of Exeter Medical School, University of Exeter, Exeter, UK. [29]Lothian Birth Cohorts, Department of Psychology, University of Edinburgh, Edinburgh, UK. [30]Alzheimer Scotland Dementia Research Centre, University of Edinburgh, Edinburgh, UK. [31]Neuroprogressive and Dementia Network, NHS Research Scotland, Scotland, UK. [32]Center for Biomarker Research and Precision Medicine (BPM), Virginia Commonwealth University, Virginia, USA. [33]Department of Psychiatry, Amsterdam UMC, Vrije Universiteit Amsterdam, Amsterdam, The Netherlands. [34]Department of Psychiatry, University of Oxford, Oxford, UK. ✉e-mail: andrew.mcintosh@ed.ac.uk

