## [Peer Review file · Nature Communications]

Insights from a Methylome-Wide Association Study of Antidepressant Exposure

Corresponding Author: Professor Andrew McIntosh

Version 0:

Reviewer comments:

Reviewer #2

(Remarks to the Author)

The authors investigated genome-wide DNA methylation patterns in response to antidepressants. This study is an important and significant to the field because it sheds light on the epigenetic effects on antidepressants exposure and genes that are involved in the process. While there are other similar studies, this study uses a larger cohort than previously reported and focuses on the active treatment period to enable a more reliable period of antidepressant exposure. Overall, the methodology is sound and largely supports the data interpretation and conclusions. Although the study is highly informative, there are issues with clarity of the information presented that requires attention.

Abstract

The abstract can be improved to better highlight the novelty of this study compared to previous studies (i.e. focus on active treatment period, etc.).

Results

(1) The authors provided the demographics of the cohort in the study but provided little description of this in the text to help focus on the main takeaways from Table 1. In addition, could the authors provide statistical analysis to show if there is significant difference between unexposed and exposed group in the prescription-derived phenotype and self-reported phenotype for then variables described in Table 1.

(2) Could the authors please provide better description regarding what "ST" and "SF" mean in the text? There are lots of these mentioned in the results section (e.g. ST10, SF16, SF18, etc.) but with little description about what they are in the material and methods and results sections. This makes it difficult to appreciate how they are tied to the results. As a suggestion, a supplementary table describing these would be very helpful for the reader.

(3) Regarding Table 2, it is unclear how this data is connected to the study. For example, did the authors search through the EWAS catalogue to determine if their current hits are had been replicated in other EWAS studies? If so, why was this not performed for the prescription-derived MWAS data? A better description of how the EWAS search is connected to the current study would provide greater clarity and systematic connection for readers.

(4) On a related note, could the authors please provide description regarding what "SR" and "PD" mean that is mentioned in the heading of the table. This would provide greater clarity for the reader.

(5) Regarding the full samples shown in Figure 2C, could the authors please provide description regarding what psychiatric disorders is being captured by this cohort in addition to MDD?

(6) Given that MDD often comorbid with other psychiatric disorders, can the authors please provide clarification if they had screened for patients with only MDD or if the MDD-only group contains patients with MDD and other diagnosis? If the latter, could the authors please change the term "MDD only" to another description that better represents the subsamples?

(7) Regarding this statement, "Notably, restricting the analyses to MDD cases did not attenuate the effect sizes of the significant CpGs (Figure 2C)", the effect sizes look visibly different for some of the probes especially for the self-report group. Can the authors better substantiate their statement by other visualizations or perform statistical comparison between All and

MDD only groups to support this claim?

Discussion

For both the differentially methylated cytosine and differentially methylated region analysis, it is curious why the MDD only prescription derived cohort did not show any significant association compared to the self-report cohort. This is surprising because narrowing the analysis to the active treatment period should potentially reduce variance in the data by limiting the effects of low adherence to the prescription. Could the authors provide some commentary on this issue in the discussion?

(Remarks on code availability)

The code provides a readme file describing functions of the scripts. The scripts also provide sufficient annotation to enable readers to work through the scripts.

Reviewer #3

(Remarks to the Author)

Manuscript #: NCOMMS-24-30594-T

Title: Antidepressant Exposure and DNA Methylation: Insights from a Methylome-Wide Association Study

Combined reviewer's report:

This is a very interesting study investigating the association between whole-blood DNA methylation and antidepressant exposure in a large sample from Generation Scotland. The study identifies eight CpGs associated with antidepressant use, demonstrating robust correlations between self-reported and prescription-derived antidepressant exposure measures. Notably, two CpGs, cg15071067 (DQUOK-AS1) and cg26277237 (KANK1), showed a significant relationship between DNA methylation and the duration of antidepressant treatment. A methylation profile score showed a significant association with antidepressant exposure in a meta-analysis of eight external datasets.

Overall, the results are very interesting, but the discussion of the main findings is superficial and could be significantly improved in a revised version of the manuscript. The authors have a substantial amount of data in the supplementary material, and some of it should be brought into the main text along with a better discussion of the data.

Main considerations for a revised version of the manuscript:

1. In Fig. 1 – Schematic of study design, it is not clear what the second panel, labeled "Data (from top to bottom)," refers to. What does "time" mean in this context? What does the cross on the x-axis mean? I think it would be beneficial to clarify these points, simplify the diagram, or consider removing it.
2. In the Introduction section, the authors mention the estimated active treatment periods (lines 154-155). Please explain how this estimated period relates to adherence to antidepressant treatment.
3. Since the authors mentioned that the self-report measures may be unreliable due to different reasons, why the methylation profile score (MPS) was trained in self-report antidepressant exposure instead of antidepressant prescription use?
4. In the results, the authors mention that in the MDD-only self-report MWAS, only cg08527546 exhibited a significant association with antidepressant exposure. However, this data is not shown in Figure 2C. Is it possible to include this data in the figure?
5. In the results, considering that no CpGs were significantly associated in the MDD-only group, no DMRs were identified, and the gene set showed no significant enrichment in the prescription-derived MWAS, could you discuss how these results impact the overall interpretation and implications of your study's analysis compared to Barbu et al. (2022)? What new insights have been gained?
6. In the Function Annotation, how the authors interpret the significant enrichment of the self-report gene set for genes expressed in the amygdala and the "synaptic vesicle membrane" ontology, while the prescription-derived gene set showed no significant enrichment? Could the difference in enrichment results between the self-report and prescription-derived measures provide insights into the underlying biological mechanisms, and what further analyses could help clarify these findings? The authors should further discuss the possible functional relevance of this finding, since there is a lot of evidence showing that antidepressant drugs change amygdala response to stress through changes in neuroplasticity. E.g.: *Transl Psychiatry* 6, e727 (2016); <https://doi.org/10.1073/pnas.1606671113>
7. In the Enrichment Analysis, the authors mentioned that all participants had a recent (~6 months) MDD DSM-IV diagnosis. Why was this time point chosen? Is it the same for both cohorts, GS and NESDA? This should be better explained in the main text. Additionally, the authors should provide a figure or table to show these results, as there is no reference to it in the main text. Also, it is important to differentiate if this is the first MDD diagnosis or a recurrent depressive episode.
8. Figure 3C is not referenced in the main text. Please add a reference to it and clarify its relevance.
9. Although this was not the primary aim of the paper, the DNAm findings could be strengthened by functional validation. For example, the DNAm described in KANK1 could be further explored in samples from the same cohort to assess possible corresponding changes in mRNA levels and protein activity (e.g. since KANK1 inhibits neurite outgrowth by interacting with components of the cytoskeleton and signaling pathways involving RhoA activity and PI3K/Akt signalling, it could be

relevant, as a proof-of-concept, to assess mRNA and protein levels of KANK1, as well as RhoA activity and PI3K/Akt signalling, in blood cells from patients used in the study). Alternatively, simple cell culture experiments could help determining the involvement of DNAm in KANK1 with antidepressant treatment and neuroplasticity regulation.

10. While the authors address the limitations in the study design by using only patients of European ancestry, they did not address other important limitations in their design and/or analysis, such as:

- i) There is no mention of specific sex-stratified analyses or discussions on sex differences in the results section of the study (despite the detailed demographics in Table 1). It is well-described that there may be important neurobiological differences between male and female in MDD and not addressing this variable may have masked some changes in DNAm, especially considering that some of the important sex-related neurobiological differences involve the amygdala. Please see the following references: *Brain Imaging Behav.* 2023 Apr 14; 1–29. and *Neuropsychopharmacology.* 2019 Jan; 44(1): 111–128.
- ii) except from smoking, there is no mention of other potential confounding factors, such as comorbidities or concomitant treatments.

Minor points:

- some of the reviews used as references in the introduction are too old and could benefit from new information regarding the mechanism of action of antidepressants in synaptic plasticity. *Ex.:Cell.* (5):1299-1313.e19. 10.1016/j.cell.2021.01.034.
- there is no information or discussion regarding the potential effect of fast-acting antidepressants on DNAm (e.g. ketamine). Since none of the patients included in the study were under treatment with such drugs, it could be relevant to mention as a limitation that the findings of the study most likely reflect the effects of monoaminergic drugs in DNAm.

(Remarks on code availability)

Reviewer #4

(Remarks to the Author)

"I co-reviewed this manuscript with one of the reviewers who provided the listed reports. This is part of the Nature Communications initiative to facilitate training in peer review and to provide appropriate recognition for Early Career Researchers who co-review manuscripts."

(Remarks on code availability)

Version 1:

Reviewer comments:

Reviewer #2

(Remarks to the Author)

The authors have adequately addressed my concerns in their rebuttal. I have no further questions.

(Remarks on code availability)

The code provides sufficient annotation to help the reader follow the analysis

Reviewer #3

(Remarks to the Author)

The authors have addressed all the points satisfactorily.

(Remarks on code availability)

Reviewer #4

(Remarks to the Author)

(Remarks on code availability)

General response to the reviewers

We would first like to thank the reviewers for taking the time to read and critically appraise our study. The comments and expertise have all helped to improve the quality of the manuscript further. The main addition to the manuscript is sex-stratified analyses of self-reported antidepressant exposure (see general points and point-by-point responses) to our study.

Many other minor changes include: 1) Statistical analyses of the differences in demographics between antidepressant exposed and unexposed groups 2) All acronyms defined in the first instance, 3) Clarification on the use of the EWAS catalog, 4) Including information on other psychiatric diagnoses measured in the sample and subsequent renaming of MDD-only to MDD-subgroup, 5) Specific descriptions of the average increase in effect estimates for the MDD-subgroup, 6) Commentary in the discussion regarding similarities in the self-report/prescription-derived antidepressant findings, 7) More informative figure legends for Figure 1 and Figure 3, 8) Description of how active treatment periods relate to treatment adherence in prescription-derived measures, 9) Including the probe significant in self-report MDD subgroup MWAS to Figure 2C, 10) Commentary in the discussion regarding amygdala tissue enrichment and sex-specific differences, 12) Supplementary Table 19, displaying the overlapping top 1% CpGs used in the NESDA-GS enrichment test, 13) Updating references in the introduction, 14) Adding limitation of not looking at rapid-acting antidepressants.

We now detail these changes and analyses in the following point by point responses below and have also updated the relevant text and figures in the resubmitted manuscript.

General points of note for all reviewers

1) Sex-stratified analysis of self-reported antidepressant exposure

We have now run sex-stratified MWAS for the self-reported antidepressant exposure ($N_{\text{females}} = 9710$, $N_{\text{males}} = 6821$). To do this, we segregated our participants based on self-reported biological sex and re-standardised the M-values for each sex group respectively, before running the same MOA model in the main analyses (without sex as a covariate). Notably, two CpGs were significantly associated with antidepressant exposure in females, cg26277237 (b

= 0.030, $p = 1.77 \times 10^{-10}$) and cg02183564 ($b = 0.023$, $p = 4.29 \times 10^{-9}$) whilst no CpGs were significantly associated with antidepressant exposure in males (Revisions Figure 1). There was a significant positive correlation across effect estimates of all CpGs ($R = 0.039$, $p < 2.2 \times 10^{-16}$) and significantly associated CpGs ($R = 0.91$, $p = 0.0015$) (Revisions Figure 2). For the eight CpGs significantly associated with antidepressant exposure in the main analysis, we observed larger effect estimates in females than in males (Revisions Figure 3).

Revisions Figure 1: Manhattan plots of the sex-stratified MWAS of self-reported antidepressant exposure. N.B This figure is now included as Supplementary Figure 6 in the revised manuscript.

Revisions Figure 2- Correlation of effect estimates in the male-only and female-only MWAS of self-reported antidepressant exposure. Two plots are shown, one depicting all CpGs, and the other depicting the CpGs significantly associated with antidepressant exposure in the non-stratified analysis. N.B This figure is now included as Supplementary Figure 7 in the revised manuscript.

Revisions Figure 3: The effect estimates in the male-only and female-only self-report antidepressant MWAS for the CpGs which are significantly associated with antidepressant exposure in non-stratified analysis. N.B This figure is reported as Supplementary Figure 8 in the revised manuscript.

We estimated the significance of this increase by calculating Z-scores and corresponding p-values as below.

$$Z - score = \frac{\beta_{Male} - \beta_{Female}}{\sqrt{\frac{SD_{Male}^2}{N} + \frac{SD_{Female}^2}{N}}}$$

Four of the significant CpGs showed nominal significance between their effect estimates between male and female stratified analyses (cg26277237; p = 0.0081, cg02183564; p=0.0132, cg08907118; p = 0.0335, cg03222540; p = 0.0305), but this did not survive FDR-correction across all probes tested in the full model. Specific changes to the manuscript to incorporate these analyses can be found in the point-by-point responses below.

2) Update to phenotype and demographics numbers

We noticed a discrepancy between our self-report phenotype numbers (N= 16, 536, exposed = 1,508 and unexposed = 15,028) and that reported by our MWAS model, MOA (N = 16,531, exposed = 1,508 and unexposed = 15,023). Five individuals had mismatching Family IDs between the covariate file and the methylation datafile, and therefore had been discarded from the MWAS. We have updated the numbers in the manuscript to reflect this. Additionally, we noticed small errors in our sex counts in Table 2, which occurred due to human error. We have now amended the numbers and note that the percentages of male: female sex in the phenotypes are unchanged.

3) Reordering of Supplementary Tables and Figures

We noticed that the ordering of the supplementary tables and figures (Introduction, Methods, Results, Discussion) was not reflecting the manuscript ordering (Introduction, Results, Discussion, Methods). We have therefore changed the order of the supplementary tables and figures to the order in which they are mentioned in the manuscript.

Point by point responses

Reviewer #2 (Remarks to the Author):

The authors investigated genome-wide DNA methylation patterns in response to antidepressants. This study is an important and significant to the field because it sheds light on the epigenetic effects on antidepressant exposure and genes that are involved in the process. While there are other similar studies, this study uses a larger cohort than previously reported and focuses on the active treatment period to enable a more reliable period of antidepressant exposure. Overall, the methodology is sound and largely supports the data interpretation and conclusions. Although the study is highly informative, there are issues with clarity of the information presented that requires attention.

Comment 1:

The abstract can be improved to better highlight the novelty of this study compared to previous studies (i.e. focus on active treatment period, etc.).

Response to Comment 1:

We thank the reviewer for their suggestion, we have now included an explicit mention of the prescription derived measures in the abstract alongside further clarification that this is the largest study to date focusing on antidepressant exposure and DNA methylation, which we hope will better highlight the novelty of this study.

Manuscript, Abstract, Page 3: *“This study tests the association of whole-blood DNA methylation and antidepressant exposure in 16,531 individuals from Generation Scotland (GS), using self-report and novel prescription-derived measures. We identified 8 associations (7 novel), and a high concordance of results between self-report and prescription derived measures. Sex-stratified analyses observed nominally significant increased effect estimates in females for four CpGs. There was observed enrichment for genes expressed in the Amygdala and annotated to synaptic vesicle membrane ontology. Two CpGs (cg15071067; DGUOK-AS1 and cg26277237; KANK1) showed correlation between DNA methylation with the time in treatment. There was a significant overlap in the top 1% of CpGs with another independent methylome-wide association study of antidepressant exposure. Finally, a methylation profile score trained on this sample showed a significant association with antidepressant exposure in a meta-analysis of eight independent external datasets. In the largest investigation to date of antidepressant exposure and DNA methylation, we demonstrate robust associations which warrant further investigation to inform on the design of more effective and tolerated treatments for depression.”*

Comment 2:

The authors provided the demographics of the cohort in the study but provided little description of this in the text to help focus on the main takeaways from Table 1. In addition, could the authors provide statistical analysis to show if there is significant difference between unexposed and exposed group in the prescription-derived phenotype and self-reported phenotype for then variables described in Table 1.

Response to Comment 2:

We thank the reviewer for this suggestion. We have performed Welch's independent samples t-test to assess the difference in continuous variables (age & BMI) and chi-squared test to assess the differences in observed and expected counts of categorical variables between the exposed and unexposed groups (for both self-report and prescription-derived measures). We find the same patterns across both the self-report and prescription-derived measures of antidepressant exposure. The antidepressant exposed group was significantly older and had significantly higher BMI than the unexposed group. Additionally, the antidepressant exposed group had a significantly different proportions of current/former/never smokers, a significantly higher proportion of females and a higher proportion of those with lifetime MDD than the unexposed group. We have summarised the results from our statistical tests Revisions Table 1 & Revisions Table 2 and have also added this information to the manuscript, which we hope will be informative to the reader.

Manuscript, Results, Page 9-10: *“To assess demographic differences between the exposed and unexposed group, we used Welch's independent samples t-tests for continuous variables (age, BMI) and chi-squared tests for categorical variables (sex, smoking status and lifetime MDD status). In both self-report and prescription-derived measures, the antidepressant exposed group were significantly older ($t_{self-report}(1970)=10.6$, $p_{self-report}=1.01 \times 10^{-25}$, $t_{prescription-derived}(1180)=7.35$, $p_{prescription-derived}=3.63 \times 10^{-13}$) and had significantly higher BMI measurements ($t_{self-report}(1700)=11.4$, $p_{self-report}=5.45 \times 10^{-29}$, $t_{prescription-derived}(980)=9.82$, $p_{prescription-derived}=8.69 \times 10^{-22}$). Additionally, both phenotypes showed that the antidepressant-exposed group had significantly different proportions of current, former and never smokers ($\chi^2_{self-report}(2)=160$, $p_{self-report}=1.69 \times 10^{-35}$, $\chi^2_{prescription-derived}(2)=114$, $p_{prescription-derived}=1.49 \times 10^{-25}$), a significantly higher proportion of females ($\chi^2_{self-report}(1)=214$, $p_{self-report}=1.53 \times 10^{-48}$,*

$\lambda^2_{\text{prescription-derived}}(1)=129$, $p_{\text{prescription-derived}}=7.62 \times 10^{-30}$) and significantly higher proportion of those with lifetime-MDD ($\lambda^2_{\text{self-report}}(1)=2170$, $p_{\text{self-report}} < 1 \times 10^{-320}$, $\lambda^2_{\text{prescription-derived}}(1)=1450$, $p_{\text{prescription-derived}}=2.72 \times 10^{-318}$).”

Revisions Table 1: Results from the Welch’s independent samples t-test, performed to assess the difference in continuous variables (age, BMI and smoking pack years) between the antidepressant exposed and unexposed groups.

Antidepressant exposure phenotype	Variable	T-statistic	Degrees of Freedom	P-value	Cohen's D
Self-report	Age	10.6	1970	1.01×10^{-25}	0.248
	BMI	11.4	1700	5.45×10^{-29}	0.36
	pack years	11.2	1660	5.37×10^{-28}	0.371
Prescription-derived	Age	7.35	1180	3.63×10^{-13}	0.231
	BMI	9.82	980	8.69×10^{-22}	0.431
	pack years	9.15	941	3.47×10^{-19}	0.416

Revisions Table 2: Results from the chi-squared tests, performed to assess the differences in categorical variables (Sex, smoking-status and MDD-status) between the antidepressant exposed and unexposed group.

Antidepressant exposure phenotype	Demographic Variable	λ^2	Degrees of Freedom	P-value
Self-report	Sex	214	1	1.53×10^{-48}
	Smoking status	160	2	1.69×10^{-35}
	Lifetime MDD	2170	1	$< 1 \times 10^{-320}$
Prescription-derived	Sex	129	1	7.62×10^{-30}
	Smoking status	114	2	1.49×10^{-25}

	Lifetime MDD	1450	1	2.72×10^{-318}
--	-----------------	------	---	-------------------------

Comment 3:

Could the authors please provide better description regarding what “ST” and “SF” mean in the text? There are lots of these mentioned in the results section (e.g. ST10, SF16, SF18, etc.) but with little description about what they are in the material and methods and results sections. This makes it difficult to appreciate how they are tied to the results. As a suggestion, a supplementary table describing these would be very helpful for the reader.

Response to Comment 3:

We appreciate this suggestion from the reviewer. We have now clearly defined these abbreviations for the Supplementary Tables (ST), Supplementary Figures (SF) and Supplementary Information (SI) in their first instance in the manuscript. Additionally, we have created additional supplementary table labelled “acronyms”, placed before the contents with acronyms as suggested. We hope this makes reading the manuscript clearer for the reader.

Manuscript, Results, Page 13: *“Notably, all participants in this cohort had a recent (~ 6 months) MDD DSM-IV diagnosis, obtained using the Composite International Diagnostic Interview, including single-episode and recurrent MDD (Supplementary Information: SI).”*

Manuscript, Results, Page 10: *“The self-report MWAS (Figure 2A, Table 2) and prescription-derived MWAS (Figure 2B, Supplementary Table 1:ST1) identified seven and four hypermethylated CpGs respectively, in those exposed to antidepressants (Supplementary Figures 1-2: SF1-2).”*

Comment 4:

Regarding Table 2, it is unclear how this data is connected to the study. For example, did the authors search through the EWAS catalogue to determine if their current hits are had been replicated in other EWAS studies? If so, why was this not performed for the prescription-derived MWAS data? A better description of how the EWAS search is connected to the

current study would provide greater clarity and systematic connection for readers.

Response to Comment 4:

We thank the reviewer for their question. This search was performed to gain a wider understanding of the significant CpGs, by assessing which traits they have been significantly associated with in previous studies (if any). We have now added this information to the manuscript. Table 2 includes information on all the CpGs significantly associated with antidepressant exposure from the prescription-derived and self-reported MWAS (n = 8). One CpG (cg01964004) is significant in the prescription-derived MWAS only, three are significant in both the MWAS' (cg04173586, cg04215689 and cg2677237), and four are significant in the self-report MWAS only (cg02183564, cg03222540, cg08907118 and cg15071067). We have now added a column to Table 2 ("Significant in SR/PD MWAS") to make this clearer.

Manuscript, Methods, Page 24-25: *"For any significant CpGs, we searched the EWAS catalog to assess their associations with other traits in the literature."*

Revisions Table 3: Revised version of Table 2 of the manuscript, which has an additional column denoting whether the CpG was significantly associated with self-reported antidepressant exposure, prescription-derived antidepressant exposure or both.

Comment 5:

On a related note, could the authors please provide description regarding what "SR" and "PD" mean that is mentioned in the heading of the table. This would provide greater clarity for the reader.

Response to Comment 5:

We thank the reviewer for their suggestion, we have now added in this clarification in the table legend.

Manuscript, Tables, Page 40-41: *"Table 2 | Eight CpGs associated with self-reported (SR) and/or prescription-derived (PD) antidepressant use. The effect estimates (β), standard-errors (SE) and P-values (P) for the self-report (SR) and prescription-derived (PD) MWAS. The EWAS catalog was searched using the ewascalog R package¹ for other studies (n > 1000)*

which report a significant CpG-trait association, accessed on 17/03/2024.”

Comment 6:

Regarding the full samples shown in Figure 2C, could the authors please provide description regarding what psychiatric disorders is being captured by this cohort in addition to MDD?

Response to Comment 6:

We thank the reviewer for their insight. The panel in Figure 2C labelled “All” shows the CpGs significantly associated with antidepressant exposure in the MWAS of all individuals in Generation Scotland. As Generation Scotland is a population-based cohort, it is likely to capture various other psychiatric disorders which are co-morbid with Major Depressive Disorder (MDD) in the population. Comprehensive screening for psychiatric disorders in Generation Scotland is currently limited to a subset of the Structured Clinical Interview for DSM-IV Non-Patient Version (SCID) (n = 19,968), which assessed single-episode MDD, recurrent MDD, and bipolar disorder (BPD). There were 52 individuals in the self-report antidepressant group with bipolar diagnoses from SCID, 23 were classed as exposed to antidepressants and 29 unexposed. Additionally, there were 18 individuals in the prescription-derived antidepressant group with bipolar diagnoses, with 9 exposed and not exposed to antidepressants. The number of individuals per SCID category (No major disorder, BPD, single-episode MDD and recurrent MDD) for each phenotype has been added to Table 1, which we hope will be informative for the reader. More details about Generation Scotland cohort are available elsewhere: Smith, B.H., Campbell, A., Linksted, P., Fitzpatrick, B., Jackson, C., Kerr, S.M., Deary, I.J., MacIntyre, D.J., Campbell, H., McGilchrist, M. and Hocking, L.J., 2013. Cohort Profile: Generation Scotland: Scottish Family Health Study (GS: SFHS). The study, its participants and their potential for genetic research on health and illness. *International journal of epidemiology*, 42(3), pp.689-700.

Manuscript, Tables, Page 38: “*Table 1 | Demographics and SCID diagnoses of antidepressant exposed and unexposed individuals using the prescription-derived and self-reported antidepressant exposure phenotypes in Generation Scotland. M = Mean, SD = Standard Deviation, MDD = Major Depressive Disorder.*”

Comment 7:

Given that MDD often comorbid with other psychiatric disorders, can the authors please provide clarification if they had screened for patients with only MDD or if the MDD-only group contains patients with MDD and other diagnosis? If the latter, could the authors please change the term “MDD only” to another description that better represents the subsamples?

Response to Comment 7:

We thank the reviewer for their question. The MDD-only group was formed by filtering participants to those who received a single-episode MDD diagnosis or recurrent MDD diagnosis according to the SCID questionnaire (removing those with BPD diagnoses). Outside of the SCID questionnaire, Generation Scotland does not have a comprehensive screening platform for other psychiatric disorders, so it is possible that other conditions are present within this cohort although individuals with major psychiatric disorders are greatly under-represented in population-based studies. To reflect this, we have now changed this label to MDD-subgroup throughout the manuscript, as well as changing the labels in Supplementary Figure 4-5.

Comment 8:

Regarding this statement, “Notably, restricting the analyses to MDD cases did not attenuate the effect sizes of the significant CpGs (Figure 2C)”, the effect sizes look visibly different for some of the probes especially for the self-report group. Can the authors better substantiate their statement by other visualizations or perform statistical comparison between All and MDD only groups to support this claim?

Response to Comment 8:

We appreciate this suggestion from the reviewer. The correlation of effect sizes for all the CpGs from the MWAS on all individuals with the MWAS on the MDD subgroup are shown in Supplementary Figure 5. We have now included two further subplots of this figure showing only the significant CpGs that are displayed in Figure 2C. Both prescription derived and self-report phenotypes had a significant correlation, indicating overlapping signal ($R_{\text{self-report}} = 0.57$, $p_{\text{self-report}} < 2.2 \times 10^{-16}$, $R_{\text{prescription}} = 0.43$, $p_{\text{prescription}} < 2.2 \times 10^{-16}$). However, it is noted that the MDD-subgroup estimates tend to be larger than those derived from all-individuals MWAS (visualised by the $y=x$ line in Revisions Figure 4, now included in the manuscript as Supplementary Figure 5). Specifically, there was an average fold increase of 2.5 and 2.1 for significant CpGs (identified in either analysis) in the prescription-derived and self-report

MWAS when restricting to the MDD subgroup. We have now amended the main text in the manuscript to describe these findings, which we hope will be clearer to the reader.

Manuscript, Results, Line 254-258: “For both phenotypes, there was a significant correlation between CpG effect estimates in the full and MDD-subgroup analyses ($R_{self-report} = 0.57$, $p_{self-report} < 2.2 \times 10^{-16}$, $R_{prescription} = 0.43$, $p_{prescription} < 2.2 \times 10^{-16}$) (SF5). Notably, restricting the analyses to MDD cases resulted in an average 2.5-fold and 2.1-fold increase in the self-report and prescription-derived effect sizes of the significant CpGs, respectively (Figure 2C).”

Revisions Figure 4: The correlation of all probe effect sizes for the prescription-derived (left) and self-report (right) MWAS conducted on all individuals (x-axis) and on a subset of individuals with lifetime MDD status (y-axis). The bottom panel shows only the CpGs with a significant association in any self-report/prescription derived MWAS performed. R = Pearson Correlation Coefficient. N.B This figure is included in the manuscript as Supplementary Figure 1:

Comment 9:

For both the differentially methylated cytosine and differentially methylated region analysis, it is curious why the MDD only prescription derived cohort did not show any significant association compared to the self-report cohort. This is surprising because narrowing the analysis to the active treatment period should potentially reduce variance in the data by limiting the effects of low adherence to the prescription. Could the authors provide some commentary on this issue in the discussion?

Response to Comment 9:

We agree with the reviewer observation. However, our results demonstrate that prescription-derived measures of antidepressant exposure, based on active treatment periods, are similar to self-report measures with respect to their associations with DNAm. The more statistically significant signals from the self-report findings in this case is likely due to the larger sample size (+ 8580 individuals) compared to the prescription-derived analysis, facilitating larger power to detect associations. Firstly, for participants with self-report and prescription-level data, our groupings were largely concordant (Supplementary Figure 49). Secondly, we find a significant correlation between the effect sizes of CpGs from the self-report and prescription-derived antidepressant exposure (Supplementary Figure 5). We have now added a commentary about the difference in self-report and prescription-derived findings, as well as the implications of this to the discussion.

Manuscript, Discussion, Page 19: *“Our results show broadly consistent findings between self-report and prescription-derived measures. However, self-report measures showed stronger signal in downstream functional annotation analyses. Due to the strong correlation between effect estimates and several overlapping significant signals, this is likely due to the increased sample size and power in the self-report cohort (+ 8580 individuals).”*

Manuscript, Discussion, Page 21: *“Self-reported measures are often cheaper and easier to obtain in large-scale cohort studies⁴. Equally, the methods used in this study to derive of medication exposure using prescription records could enable passive data collection, enabling more generalisable analyses on whole populations outside of biases which influence participation in biobank cohorts⁵.”*

Reviewer #2 (Remarks on code availability):

The code provides a readme file describing functions of the scripts. The scripts also provide sufficient annotation to enable readers to work through the scripts.

Reviewer 3

General Comments:

This is a very interesting study investigating the association between whole-blood DNA methylation and antidepressant exposure in a large sample from Generation Scotland. The study identifies eight CpGs associated with antidepressant use, demonstrating robust correlations between self-reported and prescription-derived antidepressant exposure measures. Notably, two CpGs, cg15071067 (DGUOK-AS1) and cg26277237 (KANK1), showed a significant relationship between DNA methylation and the duration of antidepressant treatment. A methylation profile score showed a significant association with antidepressant exposure in a meta-analysis of eight external datasets.

Overall, the results are very interesting, but the discussion of the main findings is superficial and could be significantly improved in a revised version of the manuscript. The authors have a substantial amount of data in the supplementary material, and some of it should be brought into the main text along with a better discussion of the data.

Response to general comments:

We thank the reviewers for their interest in our findings and their comments on the discussion. We have now enhanced our discussion regarding the differences in self-report and prescription-derived findings and the significance of the significant enrichment of self-reported antidepressant exposure in the amygdala. We have also incorporated more suggestions for further analyses, such as functional validation using multi-omics data and in-vitro experiments.

Main considerations for a revised version of the manuscript:

Comment 1:

In Fig. 1 – Schematic of study design, it is not clear what the second panel, labeled "Data (from top to bottom)," refers to. What does "time" mean in this context? What does the case

with the cross on the x-axis mean? I think it would be beneficial to clarify these points, simplify the diagram, or consider removing it.

Response to Comment 1:

We appreciate the reviewers' suggestion. We have made a few improvements to the figure (headings for all panels, and 'case' changed to antidepressant exposed) and a more detailed legend for clarity.

Manuscript, Figure 1, Introduction, Page 8-9: *“Figure 1 | Schematic of study design. Data: Participants in Generation Scotland (GS) provided blood samples from which DNA*

methylation was measured. Their antidepressant exposure status was measured using both self-report questionnaires and prescription-derived measures. **Prescription-derived measures:** Repeated regular prescriptions over time (X axis) for antidepressants (purple bars) are merged to form active antidepressant treatment periods (blue bars). For individuals in an active treatment period at the time of blood sample (black cross) are classed as antidepressant exposed. **Methylome-wide association studies:** An MWAS and subsequent regional analysis and functional annotation was performed for both measures of antidepressant exposure. Additionally, an enrichment analysis was done using MBD-Sequencing data in NESDA for the self-report antidepressant exposure. **Methylation Profile Score:** Weights for a methylation profile score (MPS) of self-reported antidepressant exposure was calculated in GS using a LASSO model. Eight independent datasets then tested the association of this MPS with self-reported antidepressant exposure.”

Comment 2:

In the Introduction section, the authors mention the estimated active treatment periods (lines 154-155). Please explain how this estimated period relates to adherence to antidepressant treatment.

Response 2:

We thank the reviewer for their question and acknowledge that this information should be explicitly mentioned in the main text. For each prescription dispensed to an individual, we calculated its theoretical length from the number of tablets and instructed dosage (assuming adherence). If another prescription was filled around the same time (allowing for a 10% margin for occasional missed doses), we anticipated that the previous prescription would be nearing its expiration. In such cases, we merged these prescription events into a single active treatment period. Therefore, the active treatment periods are defined by prescription records matching our expected timeline when we assume adherence to the medication. It is possible that there could be participants who are actively picking up their prescriptions at regular intervals and not taking them, but we think it is reasonable to assume this to be unlikely. We have now added more clarification on this in the introduction:

Manuscript, Introduction, Page 5: “The calculation of active treatment periods from consecutive prescribing events could provide a potentially more reliable identification of antidepressant exposure⁶. Here, adherence to the medication is assumed given regular

prescription dispensations at an expected frequency (given their amount and dosage), rather than a singular prescribing event.”

Comment 3:

Since the authors mentioned that the self-report measures may be unreliable due to different reasons, why the methylation profile score (MPS) was trained in self-report antidepressant exposure instead of antidepressant prescription use?

Response to Comment 3: We thank the reviewer for their question. We chose to train the methylation profile score using self-reported antidepressant exposure due to the larger sample size, which provided increased statistical power. While we acknowledged that self-reported measures can be subject to recall bias, limited understanding of medication categories, and potential non-disclosure, our findings suggest a high level of concordance between self-reported and prescription-derived measures of antidepressant exposure in this study. Firstly, participants with both self-report and prescription data showed strong concordance between these phenotypes (Supplementary Figure 49). Secondly, there was a significant correlation between the effect estimates predicted for CpGs in the self-report and prescription-derived MWAS (Supplementary Figure 5). Additionally, self-reported antidepressant exposure is more frequently available in external datasets. By using self-reported measures in this analysis, we ensured consistency with the independent datasets in which we aimed to test the MPS, thereby improving the comparability and generalizability of our results.

Comment 4:

In the results, the authors mention that in the MDD-only self-report MWAS, only cg08527546 exhibited a significant association with antidepressant exposure. However, this data is not shown in Figure 2C. Is it possible to include this data in the figure?

Response 4: We thank the reviewer for their observation and have now included this probe's effect sizes in the self-report and prescription derived MWAS (all individuals) in Figure 2C. This section is also displayed in Revisions Figure 5.

C

Revisions Figure 5: A revised version of Figure 2C which includes the probe significantly associated with self-reported antidepressant exposure in MDD cases only (cg08527546).

Manuscript, Results, Page 11: “*Figure 2: Methylo-me-wide association study of self-reported (A) and prescription-derived (B) antidepressant exposure. (C) The standardised effect sizes and 95% confidence-intervals for associated CpGs ($p < 9.42 \times 10^{-8}$) for the full sample (dark-green) and MDD-subgroup sample (light-green). Effect sizes represent a per-1 standard-deviation increase in CpG methylation M-values.*”

Comment 5

In the results, considering that no CpGs were significantly associated in the MDD-only group, no DMRs were identified, and the gene set showed no significant enrichment in the prescription-derived MWAS, could you discuss how these results impact the overall interpretation and implications of your study's analysis compared to Barbu et al. (2022)? What new insights have been gained?

Response to Comment 5:

We thank the reviewer for their question. Firstly, the study has a much larger sample size (for both self-report and prescription-derived measures) than Barbu et al (2022), allowing for higher confidence in our findings. Additionally, Barbu et al (2022) did not conduct out of sample classification in external cohorts. Here, the application of the methylation profile score trained in Generation Scotland in other external datasets adds to the robustness of our findings. We additionally find that self-report and prescription-derived measures of

antidepressant exposure using our methodology of active treatment periods are largely concordant. We think this is an important observation for the scientific community. Self-reported medication information requires active participation from individuals and therefore likely to be more costly and time consuming. The calculation of active treatment periods and medication exposure from prescription records offers a passive way of data collection which could enable more generalisable analyses on whole populations (rather than biobanks which are often biased towards white and affluent individuals). Equally, it is reassuring to know that for cohorts who do not have infrastructure yet in place for record-linkage, that self-report is a largely a good measure of antidepressant exposure. We have now discussed these points further in the discussion:

Manuscript, Discussion, Page 21: *“There are several strengths of this study. The comparison of self-report and prescription-derived measures is valuable to the research community. Self-reported measures are often cheaper and easier to obtain in large-scale cohort studies⁴. Equally, the methods used in this study to derive of medication exposure using prescription records could enable passive data collection, enabling more generalisable analyses on whole populations outside of biases which influence participation in biobank cohorts⁵. Furthermore, the MDD-subgroup analysis indicates that the hypermethylation associated with antidepressant use is not primarily driven by MDD indication. Additionally, the performance of the GS-trained MPS in discriminating antidepressant exposure across eight external datasets, alongside the significant enrichment of top findings with an independent MWAS, demonstrates that this may be a generalisable biomarker indicative of antidepressant exposure.”*

Comment 6:

In the Function Annotation, how the authors interpret the significant enrichment of the self-report gene set for genes expressed in the amygdala and the "synaptic vesicle membrane" ontology, while the prescription-derived gene set showed no significant enrichment? Could the difference in enrichment results between the self-report and prescription-derived measures provide insights into the underlying biological mechanisms, and what further analyses could help clarify these findings? The authors should further discuss the possible functional relevance of this finding, since there is a lot of evidence showing that

antidepressant drugs change amygdala response to stress through changes in neuroplasticity. E.g.: *Transl Psychiatry* 6, e727 (2016); <https://doi.org/10.1073/pnas.1606671113>

Response to Comment 6: We thank the reviewer for their question and suggestions. Due to the overlapping signals, we see between the self-report and prescription-derived MWAS, we believe that the differences in gene-set enrichment is mainly due to the difference in sample-size between the two cohorts. We have now explicitly stated this in the discussion. We thank the reviewer for their suggestions about the functional relevance of the amygdala enrichment. We now discuss these findings in more depth in the manuscript.

Manuscript, Discussion, Page 19-20: *“The top CpGs in the self-report MWAS were significantly enriched for genes expressed in the amygdala, an important component of emotional brain circuits, specifically in regulating fear and stress responses⁴⁸. Genomic studies of MDD have shown enrichment in neural synaptic pathways⁴⁹ and brain regions in the meso-limbic system, including the prefrontal cortex, nucleus accumbens, hippocampus and the amygdala⁵⁰. Several meta-analyses have found evidence of amygdala hyperreactivity in those with MDD^{51–53}, while other studies have demonstrated the amygdala response to stress can be dampened through neuroplastic processes following antidepressant treatment⁵⁴ or cognitive behavioural therapy⁵⁵. Furthermore, a recent systematic review reported sex-specific differences in amygdala activity and grey matter volume in MDD¹⁶. Notably, our sex-stratified analyses found nominally significant sex differences in the DNAm-antidepressant exposure associations at four significant CpGs, with larger effects observed in females. Although there is no clear consensus regarding sex differences in antidepressant efficacy, reports have found that women generally respond better to selective serotonin reuptake inhibitors (SSRIs) than men. The nominally significant sex-differences in DNAm associations with antidepressant exposure and the significant enrichment of genes in the amygdala observed in this study highlight the potential role of the amygdala in mediating sex-specific responses to antidepressants. Future studies could investigate sex-specific DNAm-profiles of antidepressant exposure and their functional impact on the amygdala using functional imaging data.”*

Comment 7:

In the Enrichment Analysis, the authors mentioned that all participants had a recent (~6 months) MDD DSM-IV diagnosis. Why was this time point chosen? Is it the same for both cohorts, GS and NESDA? This should be better explained in the main text. Additionally, the

authors should provide a figure or table to show these results, as there is no reference to it in the main text. Also, it is important to differentiate if this is the first MDD diagnosis or a recurrent depressive episode.

Response to Comment 7:

We thank the reviewer for their comments. The 6-month time period is defined by the DSM-IV as a recent MDD diagnosis and is directly measured in the Composite International Diagnostic Interview (CIDI) which was deployed in NESDA. The NESDA MDD cases also include both single-episode and recurrent MDD, which we have now specified. In Generation Scotland, MDD is assessed as a lifetime status from the SCID questionnaire. We agree that a table should be made available to see the enrichment results. We have now included a table (Supplementary Table 19) displaying the overlapping CpGs in the top 1% of each MWAS for which there was a significant enrichment and their corresponding statistics in each MWAS. We have now referenced this in the main text, and we hope it will be informative for the reader.

Manuscript, Results, Page 13: *“We tested whether top findings from our self-reported MWAS on all participants were also more likely to be among the top findings in an independent MWAS of antidepressant exposure in the Netherlands Depression and Anxiety (NESDA) cohort¹². Notably, all participants in this cohort had a recent (~ 6 months) MDD DSM-IV diagnosis, obtained using the Composite International Diagnostic Interview, including single-episode and recurrent MDD (Supplementary Information: SI)”*

Manuscript, Results, Page 14: *“Results suggested a small (OR: 1.39) but significant ($P < 0.042$) enrichment between the top 1% of results from both MWAS’ (ST19).”*

8) Figure 3C is not referenced in the main text. Please add a reference to it and clarify its relevance.

Response 8: We thank the reviewer for spotting this oversight. We have now referenced Figure 3C in the main text and clarified that it shows the results from the meta-analysis of MPS ~ antidepressant exposure in independent cohorts. We have also added more information to the figure legend.

Manuscript, Results, Page 14: *“The random-effects meta-analysis (Figure 3C) found a significant association between antidepressant exposure and the MPS (pooled β [95%CI]:*

0.196 [0.105, 0.288], $p < 1 \times 10^{-4}$), with low heterogeneity between studies (I^2 [95%CI] = 0% [0, 64.8%]) (ST23).”

Manuscript, Results, Figure 3 Legend, Page 17: “*Figure 3: Antidepressant exposure ~ MPS in external cohorts. A) The sample sizes of each dataset, B) Nagelkerke’s pseudo R^2 , C) The effect sizes and confidence intervals of MPS ~ antidepressant exposure analysis in each cohort individually (blue squares), alongside the pooled effect estimate from the random-effects meta-analysis (blue diamond). Square size = study weight.*”

Comment 9:

Although this was not the primary aim of the paper, the DNAm findings could be strengthened by functional validation. For example, the DNAm described in KANK1 could be further explored in samples from the same cohort to assess possible corresponding changes in mRNA levels and protein activity (e.g. since KANK1 inhibits neurite outgrowth by interacting with components of the cytoskeleton and signaling pathways involving RhoA activity and PI3K/Akt signalling, it could be relevant, as a proof-of-concept, to assess mRNA and protein levels of KANK1, as well as RhoA activity and PI3K/Akt signalling, in blood cells from patients used in the study). Alternatively, simple cell culture experiments could help determining the involvement of DNAm in KANK1 with antidepressant treatment and neuroplasticity regulation.

Response to Comment 9:

We thank the reviewer for their comment and think that this analysis is an excellent idea for further disentangling these associations and their functional relevance. We are actively undertaking work in this area as part of a Wellcome Mental Health award titled “Antidepressant Medications: Biology, Exposure and Response” (AMBER) which includes an aim entirely focussed on in-vitro experiments which aim to validate findings from association studies and population datasets such as these. We believe this work to be an analysis of its own right, which is outside the scope of this manuscript. More information on AMBER investigators and projects aims can be found here:

<https://www.kcl.ac.uk/research/amber-antidepressant-medications-biology-exposure-response>. We have now included this as a future direction of research in the discussion.

Manuscript, Discussion, Page 19: “*Functional validation of the associations of antidepressant exposure with DNAm at KANK1 and DGUOK-AS1 would strengthen our*

findings. The integration of additional multi-omic data alongside in-vitro experiments could further assess the impact of these associations on biologically relevant processes, such as neuroplasticity.”

Comment 10 i):

While the authors address the limitations in the study design by using only patients of European ancestry, they did not address other important limitations in their design and/or analysis, such as:

i) There is no mention of specific sex-stratified analyses or discussions on sex differences in the results section of the study (despite the detailed demographics in Table 1). It is well-described that there may be important neurobiological differences between male and female in MDD and not addressing this variable may have masked some changes in DNAm, especially considering that some of the important sex-related neurobiological differences involve the amygdala. Please see the following references: *Brain Imaging Behav.* 2023 Apr 14: 1–29. and *Neuropsychopharmacology.* 2019 Jan; 44(1): 111–128.

Response to Comment 10 i):

We thank the reviewer for their insight and suggestions. We agree that this is an important factor to consider in our analyses and have now run sex-stratified MWAS for the self-reported antidepressant exposure ($N_{\text{females}} = 9710$, $N_{\text{males}} = 6821$). Notably, two CpGs were significantly associated with antidepressant exposure in females, cg26277237 ($\beta = 0.030$, $p = 1.77 \times 10^{-10}$) and cg02183564 ($\beta = 0.023$, $p = 4.29 \times 10^{-9}$) whilst no CpGs were significantly associated with antidepressant exposure in males. Across all CpGs there was a significant positive correlation of CpG effect sizes ($R = 0.039$, $p < 2.2 \times 10^{-16}$). For the eight CpGs significantly associated with antidepressant exposure in the main analysis, we observed larger effect estimates in females than in males. We estimated the significance of this increase by calculating Z-scores and corresponding p-values as below.

$$Z - score = \frac{\beta_{\text{Male}} - \beta_{\text{Female}}}{\sqrt{\frac{SD_{\text{Male}}^2}{N} + \frac{SD_{\text{Female}}^2}{N}}}$$

Four of the significant CpGs showed nominal significance between their effect estimates between male and female stratified analyses, but this did not survive FDR-correction. We have made the following changes to the manuscript to incorporate this information:

New Figures + Tables:

Supplementary Figure 6 – Manhattan plots for the sex-stratified MWAS of self-reported antidepressant exposure (Revisions Figure 1)

Supplementary Table 3: CpGs significantly ($p < 9.42 \times 10^{-8}$) associated with self-report antidepressant exposure in females only (Revisions Figure 2)

Supplementary Table 4: The difference in effect estimates in the male-only and female-only self-report antidepressant MWAS, alongside Z-scores and p-values for the CpGs which report a nominally significant difference (Revisions Figure 3).

Manuscript, Abstract, Page 3: *“Sex-stratified analyses observed nominally significant increased effect estimates in females for four CpGs.”*

Manuscript, Introduction, Page 4: *“Additionally, sex differences in MDD risk^{13,14}, antidepressant efficacy and side-effects are well-documented¹⁵, which may reflect sex-specific differences in neuronal circuitry¹⁶. However, precise mechanisms of these differences are unclear.”*

Manuscript, Introduction, Page 6: *“Second, the MWAS analyses were restricted to MDD cases only to assess potential confounding by MDD, and sex-stratified analyses were conducted to investigate any sex-specific effects.”*

Manuscript, Results, Page 12: *“In the sex-stratified analyses, two CpGs were significantly associated with self-report antidepressant exposure in females, cg26277237 ($\beta = 0.030$, $p = 1.77 \times 10^{-10}$) and cg02183564 ($\beta = 0.023$, $p = 4.29 \times 10^{-9}$) (ST3) and there were no significant associations with antidepressant exposure in males (SF6, ST4). The effect sizes of CpGs in the male-only and female-only analyses were significantly correlated ($R = 0.039$, $p < 2.2 \times 10^{-16}$) (SF7). Of the eight CpG sites significantly associated with antidepressant exposure in the overall analysis, all demonstrated a larger effect size in females (SF7-8). Four of these sites showed nominally significant sex differences (cg26277237; $p = 0.0081$, cg02183564; $p = 0.0132$, cg08907118; $p = 0.0335$, cg03222540; $p = 0.0305$) (ST4).”*

Manuscript, Discussion, Page 18: *“Sex-stratified analyses indicated larger effect estimates in females compared to males.”*

Manuscript, Discussion, Page 19: *“Furthermore, a recent systematic review reported sex-specific differences in amygdala activity and grey matter volume in MDD¹⁶. Notably, our sex-stratified analyses found nominally significant sex differences in the DNAm-antidepressant exposure associations at four significant CpGs, with larger effects observed in females. Although there is no clear consensus regarding sex differences in antidepressant efficacy, reports have found that women generally respond better to selective serotonin reuptake inhibitors (SSRIs) than men. The nominally significant sex-differences in DNAm associations with antidepressant exposure and the significant enrichment of genes in the amygdala observed in this study highlight the potential role of the amygdala in mediating sex-specific responses to antidepressants. Future studies could investigate sex-specific DNAm-profiles of antidepressant exposure and their functional impact on the amygdala using functional imaging data.”*

Manuscript, Methods: *“For sex-stratified analyses, we divided our participants by self-reported sex (male/female) and used the same MOA model without sex as a covariate.”*

Comment 10 ii)

except from smoking, there is no mention of other potential confounding factors, such as comorbidities or concomitant treatments.

Response to Comment 10 ii)

We acknowledge the reviewer's point regarding the consideration of other potential confounding factors, such as comorbidities and concomitant treatments. However, adjusting for all possible confounders in this relationship would be extremely complex, and may introduce issues such as collinearity and collider bias into our results. Ultimately, randomisation is required to fully account for all known and unknown confounding factors of antidepressant exposure and DNA methylation. Specifically, randomised controlled trials incorporating DNA methylation measures could provide a robust method to control for these factors. We have now added this point to the discussion in the manuscript.

Manuscript, Discussion, Page 21: *“Thirdly, our epidemiological analyses do not adjust for various other potential confounders, such as comorbidities and concomitant treatments. However, adjusting for all possible confounds such as these may bias findings due to collinearity and collider bias.”*

Minor points:

Minor point 1)

some of the reviews used as references in the introduction are too old and could benefit from new information regarding the mechanism of action of antidepressants in synaptic plasticity. Ex.:Cell. (5):1299-1313.e19. 10.1016/j.cell.2021.01.034.

Response to minor point 1)

We thank the reviewer for their input and signposting us to highly relevant studies. We have replaced the following reference 5 with more up to date literature and have also now included more detail into the recent literature regarding antidepressant action on synaptic plasticity processes, specifically through the direct binding to TRKB receptor.

Old reference:

- 1) Sackeim HA. The definition and meaning of treatment-resistant depression. J Clin Psychiatry. 2001;62 Suppl 16:10-17.

New reference:

- 1) McIntyre RS, Alsuwaidan M, Baune BT, et al. Treatment-resistant depression: definition, prevalence, detection, management, and investigational interventions. World Psychiatry. 2023;22(3):394-412. doi:10.1002/wps.21120

Manuscript, Introduction, Page 3-4: *“The mechanism of currently prescribed antidepressants is incompletely understood. Initial theories surmised that their therapeutic effects were primarily due to the inhibition of monoamine reuptake in the synapse, leading to an increase in monoamine concentrations in the brain¹⁷. However, antidepressant treatment has a delayed onset for symptomatic improvement, which does not reflect the immediate effect on monoamine levels¹⁸. This casts doubt on the simple role of monoamines as a causal factor in MDD¹⁹⁻²¹, although other experimental paradigms continue to suggest their importance²². Another prominent theory suggests that antidepressants exert their therapeutic effects by*

increasing brain-derived neurotrophic factor (BDNF), leading to synaptic remodelling²³ and enhanced neuronal plasticity^{20,24}. A recent study found evidence of antidepressants binding directly to the BDNF receptor (neurotrophic tyrosine kinase receptor 2; TRKB), proposing this as a potential mechanism of action independent of changes in monoamine concentrations²⁵”

Minor point 2:

there is no information or discussion regarding the potential effect of fast-acting antidepressants on DNAm (e.g. ketamine). Since none of the patients included in the study were under treatment with such drugs, it could be relevant to mention as a limitation that the findings of the study most likely reflect the effects of monoaminergic drugs in DNAm.

Response to minor point 2:

We thank the reviewer for their comment and agree that this is a limitation to our study. We have therefore added this to our discussion section.

Manuscript, Discussion, Page 22: *“Finally, this study does not examine recently approved rapid-acting antidepressants, such as ketamine. Future research could build on this by exploring the associations between DNAm and rapid-acting antidepressants, while comparing these effects with those of slower-acting antidepressants discussed in this study.”*

Reviewer #4 (Remarks to the Author):

"I co-reviewed this manuscript with one of the reviewers who provided the listed reports. This is part of the Nature Communications initiative to facilitate training in peer review and to provide appropriate recognition for Early Career Researchers who co-review manuscripts."

We thank all the reviewers for their comments and suggestions, and we believe we present a sufficiently revised manuscript which is improved from first submission, thanks to the review process.